



# Pacific climate reflected in Waipuna Cave dripwater hydrochemistry

Cinthya Nava-Fernandez[1], Adam Hartland[2], Fernando Gázquez[3], Ola Kwiecien[1], Norbert Marwan[4], Bethany Fox[5,2], John Hellstrom[6], Andrew Pearson[2], Brittany Ward[2], Amanda French[2], David A. Hodell[7], Adrian Immenhauser[1], Sebastian F.M. Breitenbach[1]

[1]Sediment- and Isotope Geology, Institute for Geology, Mineralogy and Geophysics, Ruhr-Universität Bochum, Universitätsstr. 150, 44801 Bochum, Germany
[2]Environmental Research Institute, School of Science, Faculty of Science and Engineering, University of Waikato, Hamilton, Waikato, New Zealand
[3]Department of Biology and Geology, Universidad de Almeria, Almería, 04120, Spain
[4]Potsdam Institute for Climate Impact Research (PIK), Member of the Leibniz Association, Potsdam, Germany
[5]Department of Biological and Geographical Sciences, School of Applied Sciences, University of Huddersfield, Queensgate, Huddersfield, UK
[6]School of Earth Sciences, The University of Melbourne, Australia
[7]Godwin Laboratory for Palaeoclimate Research, Department of Earth Sciences, University of Cambridge, Downing Street, Cambridge CB2 3EQ, UK

*Correspondence to:* Cinthya Nava (cinthya.navafernandez@rub.de)

**Abstract**

Cave microclimatic and geochemical monitoring is vitally important for correct interpretations of proxy time series from speleothems with regard to past climatic and environmental dynamics. We present results of a comprehensive cave monitoring programme in Waipuna Cave in the North Island of New Zealand, a region that is strongly influenced by the southern Westerlies and the El Niño-Southern Oscillation (ENSO). This study aims to characterise the response of the Waipuna Cave hydrological system to atmospheric circulation dynamics in the southwestern Pacific region in order to secure the quality of ongoing palaeo-environmental reconstructions from this cave.

Cave air and water temperatures, drip rates, and $CO_2$, concentration were measured, and samples for water isotopes ($\delta^{18}O$, $\delta D$, d-excess, $^{17}O_{excess}$) and elemental ratios (Mg/Ca, Sr/Ca), were collected continuously and/or at monthly intervals from 10 drip sites inside Waipuna Cave for a period of ca. 3 years. These datasets were compared to surface air temperature, rainfall, and potential evaporation from nearby meteorological stations to test the degree of signal transfer and expression of surface environmental conditions in Waipuna Cave hydrochemistry.

Based on the drip response dynamics to rainfall and other characteristics we identify three hydrological pathways in Waipuna Cave: diffuse flow, combined flow, and fracture flow. Dripwater isotopes do not reflect seasonal variability, but show higher values during severe drought. Dripwater $\delta^{18}O$ values display limited variability and reflect the mean isotopic signature of precipitation, testifying to rapid and thorough buffering in the epikarst. Mg/Ca and Sr/Ca ratios in dripwaters are predominantly controlled by prior calcite precipitation (PCP). Prior calcite precipitation is strongest during austral summer (December-February), reflecting drier conditions and lack of effective infiltration, and is weakest during the wet austral winter (July–September). The Sr/Ca ratio is particularly sensitive to ENSO conditions due to the interplay of congruent/incongruent host rock dissolution, which manifests itself in lower Sr/Ca in above-average warmer and wetter (La Niña-like) conditions. Our microclimatic observations at Waipuna Cave provide valuable baseline for perceptive interpretation of speleothem proxy records aiming at reconstructing the past expression of Pacific climate modes.

## 1 Introduction



The southwestern fringe of the Pacific Ocean between 30º and 40ºS marks the transition zone between the tropical Pacific and the sub-tropical Southern Ocean. This region's position near the boundary of two markedly different climates makes the SW Pacific a key site to capture the signatures of the coupled atmospheric-ocean climate subsystems of the El Niño-Southern-Oscillation (ENSO) and the southern Westerlies (Basher, 1998; Shulmeister et al., 2004). The effects of both these circulation features are well expressed in seasonal to multi-annual climate variability in New Zealand (Mullan, 1995). New Zealand's climate is strongly modulated by both ENSO and the southern Westerlies, and its agricultural economy reacts sensitively to inter-annual fluctuations in weather patterns caused by their dynamics (Basher, 1998). During El Niño events, New Zealand is susceptible to increases in the frequency and intensity of westerly and southwesterly winds, accompanied by decreased rainfall in eastern New Zealand. In contrast, La Niña events are accompanied by stronger northeasterly winds and increased rainfall in northeast New Zealand (Griffiths, 2006). The environmental and economic costs of strong ENSO events for New Zealand are considerable; for example, the severe drought triggered by the strong El Niño event of 1997–1998 caused economic losses of ca. 1 billion NZD (Basher, 1998). The projected effects of ENSO on New Zealand hydroclimate are based on observations of El Niño and La Niña dynamics recorded over the instrumental period. However, such observations cover only a comparatively short time span. The study of the long-term natural variability of New Zealand hydroclimate (and emergent teleconnection patterns) is a priority, both, because of the effects of ENSO variability on local economic conditions and because records from this region are sparse but vital for improving the robustness of model projections of future ENSO conditions.

Speleothems (secondary cave carbonates) offer precise chronological control and a wide range of environmentally-sensitive proxies, including growth rate, carbon and oxygen isotopes ($\delta^{13}C$, $\delta^{18}O$), major and trace elements, and increasingly, non-traditional isotope systems such as $\delta^{44}Ca$ (Henderson, 2006; Fairchild and Baker 2012; Owen et al., 2016; Magiera et al.,2019) or $\delta^{26}Mg$ (Immenhauser et al., 2010; Riechelmann et al., 2012).

Over the last two decades, speleothems have provided invaluable reconstructions of past rainfall, changes in vegetation, and coupled atmospheric-ocean dynamics (Dorale et al., 1998; Asmerom et al., 2010; Myers et al., 2015; Chen et al., 2016; Griffiths et al., 2016; Lechleitner et al., 2017; Kaushal et al., 2018). Speleothem-based records are among the few climatic reconstructions that allow modern calibration, linking palaeo-data from stalagmites with meteorological and direct in-cave monitoring and making it possible to trace climatic signals from the surface to the speleothem at timescales from seasonal (Frappier et al., 2002) to orbital (Meckler et al., 2012; Mattey et al., 2008). Apart from established methods, ongoing investigations are exploring the use of triple oxygen isotopes (i.e. $^{17}O_{excess}$; see section 3.5 for definition) in carbonate and fluid inclusions in speleothems as a proxy for changes in atmospheric humidity (Affolter et al., 2015). In rainfall - and presumably in cave dripwater - $^{17}O_{excess}$ mostly depends on the relative humidity during formation of water vapour at the moisture source (i.e. ocean surface), with temperature having a minor effect (Uechi and Uemura, 2019).

Monitoring of modern cave environments, encompassing ventilation, hydrology and hydrochemistry, is critical for reliable interpretations of palaeo-environmental proxies preserved in speleothems (McDermott, 2004; Fairchild et al., 2006a; Breitenbach et al., 2015). The key parameters affecting a speleothem's fidelity as an environmental archive include cave air and water temperature, drip discharge dynamics, and cave air $p$CO$_2$, as well as dripwater chemistry (Fairchild and Baker, 2012; Tremaine et al., 2016). Analysis of the latter allows the disentangling of the processes involved in the transfer of the external environmental signals (e.g., precipitation history, temperature, or soil dynamics) from those inherent to the epikarst and cave (e.g., degree of water-rock interaction, seepage-water $CO_2$ degassing, cave air $CO_2$ dynamics and prior carbonate precipitation; Oster et al., 2012; Fairchild and Baker, 2012).

The characterization of infiltration pathways is an essential prerequisite for delineating the processes that can modulate dripwater chemistry, i.e. the degree of water-rock interaction taking place in the epikarst and the climate signal transferred by the dripwater to the speleothems. The physical propreties of the karst zone define the different hydrological pathways: i) primary porosity that promotes slow flow through a diffuse matrix; ii) fractures and fissures that allow faster water flow; and iii) conduits with high flow rates (Ford and Williams, 2007; Fairchild and Baker 2012). Conceptual models of cave dripwater





hydrology have traditionally sought to delineate these different types of flow routing on the basis of peak discharge and discharge variability Smart and Friederich (1987), later modified by Baker et al. (1997). More recently, Jex et al. (2012), Markowska et al. (2015), and Mahmud et al. (2018) proposed new classification systems based on long-term drip discharge time series, statistical tests, and clustering models. These site-specific classification systems argue that drip discharge characterisation enables a better understanding of the controls on stalagmite growth and of climate proxies such as stable

isotopes and trace metals.

Flow routing to speleothem drip points is the first-order control on dripwater hydrochemistry, with particular relevance for trace elements and other proxies of prior calcite precipitation (PCP) (Fairchild et al., 2000; Wassenburg et al., 2012). PCP serves as a proxy system for moisture availability (Magiera et al., 2019), and affects a range of trace elements, which may either become more concentrated (increasing X/Ca) or diluted in solution (decreasing X/Ca) with increasing PCP (Hartland

and Zitoun, 2018). Common PCP proxy systems include the Group II alkaline earth metals (Mg, Sr, Ba) and stable Ca isotopes ($\delta^{44}$Ca) (Magiera et al., 2019; Owen et al., 2016). Calcium isotopes show particular potential for future quantitative PCP reconstructions, however linking PCP to rainfall amount requires careful site-specific monitoring and calibration (Li et al., 2018).

Cave monitoring studies in the south-western Pacific regions have documented strong ENSO signals in Australia (Tadros et

al., 2016), Borneo (Moerman et al., 2014) and on Niue Island in the central Pacific (Tremaine et al., 2016). In New Zealand, however, the number of comparable monitoring studies is still limited. Williams and Fowler (2002) investigated the relationship between the oxygen isotope composition of rainfall and dripwater in Aranui Cave (Waitomo region, North Island). They found that neither the seasonal variability nor the ENSO-related variability detected in $\delta^{18}$O values of rainfall was transferred to the cave dripwaters. In the case of Aranui Cave, this seems to result from homogenization of the water isotope

signal on its path through the soil and epikarst (Williams and Fowler, 2002). These results highlight the importance of understanding local settings and indicate a need for revisiting other New Zealand cave systems in order to test the relationship between external environmental signals and those inherent to the epikarst and cave system.

We hypothesize the hydrochemistry of Waipuna Cave is sensitive to changes in precipitation patterns, and thus to seasonal variations and dynamics related to ENSO and the southern Westerlies, due to its geographical position and geometry. Our

study aims to test this hypothesis through a 3-year cave monitoring study, including measurements of cave ventilation, dripwater hydrochemistry, and local temperature patterns. Our study has three consecutive objectives: i) characterisation of the dripwater chemistry, including major and trace elements (Mg/Ca and Sr/Ca) and the isotope geochemistry ($\delta^{17}$O, $\delta^{18}$O, $\delta$D and d-excess); ii) identification of the mechanisms controlling dripwater chemistry; and iii) their relation to precipitation changes, seasonal and interannual (ENSO) climate variability.


## 2 Study area

### 2.1 Geographical and climatological setting

Waipuna Cave is located in the Waitomo district, North Island, New Zealand (S 38°18'41.3'', E 175°1'14.3'', 395 m above sea level), ca. 27 km from the west coast (Fig. 1a). The Waipuna Cave is a ca. 3.5 km long river cave developed in the clay-

rich, stylobedded Oligocene Pancake Limestone (Nelson, 1973, Fig. 1b). The estimated bedrock overburden is ca. 20–30 m. The main passage is accessed via a ca. 25 meter-deep doline. An underground stream flowing through the cave connects a number of larger chambers (Fig. 1c, d). The winding and narrow passage that connects the main chambers limits cave air flow, and the cave atmosphere is relatively isolated from the surface conditions (Fig. 1c). The surface morphology around Waipuna Cave is characterised by a craggy karst landscape with frequent large dolines (Fig. 1b).

The soil zone is generally > 1 m thick typic orthic allophanic (LO) developed on extensive and exceptionally well drained North Island rhyolitic volcanic ash deposits (Hewitt, 2010). The vegetation cover is a patchwork of lush podocarp-hardwood forest with a dense undergrowth of shrubs, ferns and tree-ferns. This is surrounded by grassland pasture used for grazing cattle.





According to Garr et al. (1991), the region receives a mean annual rainfall of 1407.5 mm, without any distinct rainy season. Summer and winter precipitation means are very similar (104.5 mm and 135 mm, respectively), with highest values in austral

autumn (April) and lowest values during austral summer (December). The mean annual temperature is 12.8°C, ranging from an average of 17.4°C in summer to 8°C in winter.

## 3 Methods

### 3.1 External environmental monitoring

Several meteorological datasets were used to constrain the relationship between surface and in-cave environmental conditions. A HOBO temperature logger (ONSET, Bourne, Massachusetts) with a precision of ± 0.2°C housed in a purpose-built meteorological station, was deployed a few hundred meters from the cave entrance and, recorded air temperature at half-hourly intervals between June 2017 and May 2018, from which daily means were calculated. The same station recorded daily rainfall over the periods April 2016 to September 2016 and May 2017 to May 2018. Rainfall was recorded with a precision of 0.1 mm

using a combination of Campbell-Scientific (established in September 2016) and HOBO tipping-bucket rain gauges (established May 2017). Due to technical difficulties, it was not feasible to collect rainfall data over the entire monitoring period. To complement the local meteorological dataset, daily rainfall and potential evapotranspiration (PET, based on the Priestley-Taylor equation) data were obtained from the NIWA National Database (www.clifo.niwa.co.nz) using proximal stations at Otorohanga Glenbrook and Te Kuiti Ews, 22 km and 13 km from Waipuna Cave, respectively. In addition to

meteorological monitoring, cumulative monthly rainwater samples were collected using a passive rainwater collector (IAEA, 2014).

### 3.2 Cave environment monitoring

Waipuna Cave was visited at ca. monthly intervals for a period of almost 3 years from April 2016 to February 2019 (32 visits in total). Discrete cave air $pCO_2$ measurements were conducted during each visit in the sampling chamber using a Vaisala

M170 GMP 343 Carbocap $CO_2$ probe with a precision of ± 10 %. These measurements were always conducted before all team members entered the cave chamber to avoid contamination. Water temperature, pH, and electrical conductivity were manually measured on the dripwaters using LAQUAtwin pH and conductivity probes (HORIBA Scientific Japan) calibrated prior to each sampling event. Air temperature in the Organ Loft was recorded every 30 minutes throughout the monitoring period using an automatic HOBO logger.

### 3.3 Cave water collection and drip rates

Ten drip sites in the Organ Loft chamber were sampled for water isotopes and elemental concentrations. Seven of the drip sites (WP 1-1, WP 1-2, WP 1-3, WP 1-4, WP-1A, WP 1B and WP-FB) feed a flowstone (Fig. 2a, b) (from which the WP-15-1 core was obtained). Three further drip sites (WP-2, WP-3 and WP-4) are located a few meters apart below active stalactites (Fig. 2c) on an elevated terrace within the same chamber as the first seven drip sites. Water from the cave stream was collected

during each cave visit. Dripwater samples for stable isotope analysis ($\delta^{17}O$, $\delta^{18}O$ and $\delta D$) were collected and stored in sterile 2 ml or 10 ml polypropylene bottles, filled with no head space and sealed using laboratory film. Water samples for trace element analysis were collected in 15 ml polypropylene (Falcon) tubes, previously demonstrated to have low metal blanks. These samples were acidified with 2 % $HNO_3$ (using in-house, double Teflon-distilled acid) and refrigerated until analysis.

Drip rates at the monitored sites were determined using two independent methods. For all drip sites, the number of drips per

minute were counted during each visit using a stop watch and counting at least 10 drips (normally at least three one-minute duration counts for the fast drip points). Drip rates were also automatically recorded at the four drip sites WP 1-1, WP 1-2,



WP 1-3, and WP-2 using acoustic Driptych Stalagmate drip loggers (http://www.driptych.com/). Cross correlation of cumulative antecedent rainfall and drip rates was carried out with the aim of understanding the discharge response.

The coefficient of variation CV (Smart and Friedrich, 1987; Baldini et al., 2006) was calculated for discharge relative to the time of data collection (CV % = $\sigma/\bar{u}$ x 100, with $\sigma$ being the standard deviation, and $\bar{u}$ the mean).

### 3.4 Oxygen and hydrogen isotopes of water

The oxygen and hydrogen isotope composition ($\delta^{17}O$, $\delta^{18}O$ and $\delta D$) of dripwater, and stream water was measured using cavity ring down spectroscopy (CRDS) (Steig et al., 2014). Dripwater samples collected between August 2016 and April 2017 were analysed for $\delta^{18}O$ and $\delta D$ using a Picarro L1102-i water isotope analyzer at the Godwin Laboratory for Paleoclimate Research, University of Cambridge, UK. Samples collected between June 2017 and February 2018 were analysed for $\delta^{17}O$, $\delta^{18}O$ and $\delta D$ using a Picarro L2140-i at the School of Environmental Sciences of the University of St. Andrews, UK. Samples collected from March 2018 to February 2019 were measured for $\delta^{17}O$, $\delta^{18}O$ and $\delta D$ on a Picarro L2140-i at the Department of Biology and Geology at the Universidad de Almería, Spain.

The results were normalized to the V-SMOW (Vienna-Standard Mean Ocean Water) scale by analyzing internal standards before and after each measurement set of ten to twelve samples. Three internal water standards (JRW, BOTTY and SPIT) were calibrated against V-SMOW and SLAP (Standard Light Antarctic Precipitation), using $\delta^{17}O$ values of 0.0 ‰ and -29.69865 ‰, respectively, and $\delta^{18}O$ values of 0.0 ‰ and -55.5 ‰, respectively (Schoenemann et al. 2013). This standardization considers $^{17}O_{excess}$ = 0 for both international standards. $\delta D$ was calibrated against V-SMOW, GISP (Green Ice Sheet Precipitation) and SLAP. All isotopic deviations are reported in parts per thousand (‰) relative to V-SMOW. $^{17}O_{excess}$ values are given in per meg units (0.001 ‰), where $^{17}O_{excess}$ = ln($\delta^{18}O/1000+1$)-0.528*ln($\delta^{17}O/1000+1$) (Barkan and Luz, 2005). The $^{17}O_{excess}$ expresses a small $\delta^{17}O$ deviation (normally a few per meg units) of a water sample with respect to the Global Meteoric Water Line (GMWL) for triple oxygen isotopes, for which the slope is 0.528 (Luz and Barkan, 2010). The d-excess describes the deviation for $\delta18O$ and $\delta D$ of a given sample with respect to the GMWL (dD-8*d18O; Craig et al., 1961).

The long-term precision (1$\sigma$) of oxygen and hydrogen isotope analyses was evaluated by measuring an internal standard (BOTTY) every 6 samples. The long-term precision on the Picarro L1102-i was ± 0.08 ‰ for $\delta^{18}O$ and ± 0.7 ‰ for $\delta D$ (n = 33), while on the Picarro L2140-i it was ± 0.03 ‰, ± 0.05 ‰ and ± 0.4 ‰ for $\delta^{17}O$, $\delta^{18}O$, and $\delta D$, respectively (n = 43). The long-term precision for the d-excess parameter ($\delta D-8*\delta^{18}O$) was ± 0.7 ‰ on the Picarro L1102-I and ± 0.3 ‰ on the Picarro L2140-i. The long-term precision for $^{17}O_{excess}$ was ± 8 per meg. The calibrated value of BOTTY was indistinguishable within analytical errors when using the three different instruments, suggesting results are comparable.

### 3.5 Water major and trace elements

Elemental and major cation concentrations in cave stream, drip- and rainwater were measured on two generations of instruments at the University of Waikato. Samples collected between August 2016 and October 2017 were analysed using a Perkin Elmer Elan quadrupole ICP-MS, and samples collected between November 2017 and February 2019 were analysed with an Agilent 8900 triple quadrupole ICP-MS at the Waikato Environmental Geochemistry Laboratory. The ICP-MS was optimized to maximum sensitivity daily, ensuring oxides and double-charged species were less than 2 %. External calibration standards were prepared using a IV71-A multi element standard from 0.1 to 500 ppb for trace elements and single element standards were used to prepare calibration standards for major elements Ca, Fe, Si, P, S, K, Na. An internal standard containing





Sc, Ge, Te, Ir and Rh was used for all samples. Check standards were analysed every 20 samples and re-calibration was performed every 100 samples. Blank samples were analysed every 10 samples to ensure minimal carryover between analyses.

## 4 Results

### 4.1 Local meteorology

The available daily precipitation and temperature datasets from the stations at Waipuna, Otorohanga Glenbrook and Te Kuiti Ews show the same pattern where overlapping, although the amplitudes differ (Figure 3). The Waipuna meteorological station records large variations in annual surface temperature, with minimum and maximum temperatures ranging from -0.6°C in July to 33°C in January. Precipitation ranged from 118 mm in May 2017 to 51 mm in December 2018, without any pronounced seasonality. The driest months are typically November and December (austral summer), and the wettest months are August

and September. While the variability and timing of rainfall at the three stations correspond to the same seasonal structure, the Waipuna rain station typically recorded higher amounts. This is consistent with its higher altitude (~90 m) relative to Otorohanga Glenbrook (40 m) and Te Kuiti Ews (62 m), and thus indicates an orographic effect on rainfall.

The mean monthly surface conditions (2002–2019) from Te Kuiti Ews station are illustrated in figure 4. These comprise monthly mean rainfall, monthly mean temperature, potential evapotranspiration (PET), and effective rainfall ($P_{eff}$), calculated

as the difference between P and PET.

### 4.2 Waipuna Cave hydrology

All drip sites were hydrologically active during the monitoring period, with variable mean discharges between 10.5 and 22.1 $\mu$L s$^{-1}$. Cross-correlation analysis between antecedent cumulative rainfall and drip rate time series from the acoustic drip

loggers show different lag times for each drip (Table 1, Fig. 5).  These are 19 days for WP 1-1, 15 days for WP 1-2, 16 days for WP 1-3, and 4 days for drip point WP-2 (Fig. 5).

For the drip sites where drip rates were only measured manually during the cave visits, the observed lags were 18 days for WP 1-4 and WP 1A, 11 days for WP 1B and WP-3, and 6 days for WP-4. The coefficient of variation (CV) of the drip sites varied between 31 and 149 % (Table 1). According to the classification system of Smart and Friedrich (1987), drip sites WP 1A, WP

1B and WP-3 reflect seepage flow, while sites WP 1-1, WP 1-2, WP 1-3, WP 1-4, WP FB, WP-4, and WP-2 are fed by fracture flow (Supplement S1). However, the variation in lag responses, the cluster analysis (see below) and the physical observations in the cave all indicate that this system of classification is not an ideal fit to our drip rate responses, with all drips instead probably falling somewhere along a continuum between more diffuse (seepage) and fracture dominated.

Cluster analysis reveals three main clusters of drip sites based on 25 observations of discharge at each drip site with 4 common

data points among them (Fig. 6). Drip site WP-2 is statistically different from all the others. A second cluster is formed by sites WP1-3 and WP-3 (using the combined logger and manual observations), and a third cluster comprises the drips WP 1-1, WP 1-2 (manual and logger observations), WP 1-4 and WP-4. This classification is reasonably consistent with lag in response to rainfall for each drip site, with the exception of WP-4, which has a fast response to rainfall but is grouped with drip sites with much slower responses.


### 4.3 Isotope geochemistry





Dripwater oxygen isotope values varied between -3.01 and -2.44 ‰ for $\delta^{17}O$, -5.85 and -5.20 ‰ for $\delta^{18}O$, and -32.7 and -28.3 ‰ for $\delta D$. The d-excess ranged from 8.8 to 15.4 ‰ (Supplement S6). All dripwater $\delta^{18}O$ and $\delta D$ values fall on the local meteoric water line (LMWL, $\delta D = 7.15*\delta^{18}O+7.6$) as determined from rainwater samples from the Waikato region in the North Island (Fig. 7) (Keller, 2014). A linear regression of all dripwaters is expressed as $\delta D = 5.05*\delta^{18}O-2.3$. The very low intercept of -2.3 is the result of the very narrow range of the dripwater cluster and is not related to secondary evaporation. The $\delta^{18}O$ values of the cave stream are in the same range as the dripwaters, but $\delta D$ values are ~ 20 ‰ higher ($\delta D = 5.9*\delta^{18}O+3.4$). The d-excess of the stream water ranged from 9.4 to 15 ‰.

All dripwater $\delta^{17}O$ and $\delta^{18}O$ values fall close to the GMWL (Supplement S2), the equation for which is $\ln(\delta^{17}O/1000+1) = 0.528 \ln(\delta^{18}O/1000+1) +0.00033$ (Luz and Barkan, 2010). The mean $\delta^{17}O$ deviation with respect to the GMWL ($^{17}O_{excess}$) in dripwater is 26 ± 8 per meg, and the values ranged from 6 to 44 per meg. The $^{17}O_{excess}$ values of the cave stream were 25 ± 4 per meg on average.

Drip sites WP 1-1, WP 1-2, WP 1-3, WP 1-4, WP 1A, and WP 1B show low variability in dripwater $\delta^{18}O$ and $\delta D$ (-5.5 to -5.7 ‰ for $\delta^{18}O$ and -28.5 to -32.5 ‰ for $\delta D$, which although low is still greater than the analytical error of 0.16 ‰ and 1.4 ‰, (2$\sigma$), respectively) between September 2016 and June 2017 (Fig. 8b and c). From July 2017 to January 2019 virtually no variability was observed in $\delta^{18}O$ and $\delta D$ (Supplement S3). By contrast, the three drip sites with the shortest lags, WP-2, WP-3, and WP-4, exhibit higher variability in $\delta^{18}O$ and $\delta D$. In particular, sites WP-2 and WP-3 show a marked increase (0.5 ‰) in $\delta^{18}O$ between December 2016 and January 2017 (Fig. 8a), $\delta^{17}O$ varies in the same way as $\delta^{18}O$ (Supplement S4).

### 4.4 Dripwater major and trace elements

Here we report elemental composition data for those components typically influenced by prior calcite precipitation (PCP), i.e. Ca concentration (Fairchild et al., 2000) and Mg/Ca and Sr/Ca ratios (Tremaine et al., 2016). Dripwater Ca concentrations range from 0.72 to 2.27 molL$^{-1}$, Mg/Ca varied from 17.77 to 66.55 and Sr/Ca varied from 0.24 to 1.116 (mmol mol$^{-1}$). All drip sites show a strong positive correlation between Mg/Ca and Sr/Ca ratios, with coefficients of determination varying from 0.68 to 0.90 (Fig. 9b). The two different trends observed in the relationship between Mg/Ca and Sr/Ca ratios will be discussed in section 5.5. The temporal variability of the Mg/Ca and Sr/Ca ratios shows three noticeable peaks for all drip sites during the summer (February) 2017, 2018 (January) and 2019 after the previously reduced effective rainfall. Dripwaters collected during late summer (January-February) generally had higher Mg/Ca and Sr/Ca ratios and lower Ca concentrations, while samples collected in winter (July-August) had the lowest Mg/Ca and Sr/Ca ratios and the highest Ca concentrations (Fig. 10 and Supplement S7).

### 4.5 Cave air temperature and CO$_2$

Daily cave air temperature in the Organ Loft chamber between June 2017 and May 2018 varied between 7.4°C and 11.7°C (mean 10.4°C) (Fig. 11a, b), with the highest temperatures measured in summer (February and March) and the lowest temperatures in winter (July and August). The air temperature logger at Waipuna Cave meteorological station recorded temperatures between -0.6°C and 29.3°C. The temperature difference ($\Delta T$) between Waipuna Cave air and the external air shows an annual cycle (Fig. 11c), ranging from -22°C to 8.3°C, with a mean of -2.1°C. The largest negative $\Delta T$ values occurred from late spring November 2017 to early autumn April 2017 owing to the marked increase in external temperature.

Cave air $p$CO$_2$ varied from a minimum of 438 ppm in September 2016 to a maximum of 930 ppm recorded in March 2019 (Fig. 11b). Cave air $p$CO$_2$ is positively correlated with cave air temperature ($R^2 = 0.67$, $p = 0.045$).





## 5 Discussion

This work aimed at evaluating the hydrochemical response of the Waipuna Cave as a sensor of external environmental variability and, subsequently, calibrating environmentally-sensitive proxies from speleothem archives. Here, we explore the

links between the physicochemical parameters measured in Waipuna Cave and rainfall and temperature changes at seasonal to inter-annual timescales. Our results show that Waipuna Cave reflects the external environmental dynamics on inter-annual timescales. The results and interpretation of monitoring data constitute a solid platform for the extraction of palaeoclimate information from a flowstone core (WP-15-1), currently being analysed by our research group.

### 5.1 Waipuna Cave hydrology

The meteorological data from Otorohanga Glenbrook and Te Kuiti Ews stations are considered suitable for evaluating the Waipuna Cave monitoring data, given the rainfall patterns are similar at the Waipuna. The monitored drip sites in Waipuna Cave are highly sensitive to rainfall variability, demonstrating the free draining nature of the overlying soil and the relatively thin bedrock overburden.

On the basis of the classification model of Smart and Friedrich (1987), drip sites WP-1A, WP-1B and WP-3 qualify as *seepage*

*flow,* characterised by continuous flow with low dripwater rate variability. According to the same classification, drip sites WP-1-1, WP 1-2, WP 1-3, WP 1-4, WP-FB, WP-4 and WP-2 fall into the *seasonal flow* category (Supplement S1). However, the discharge variability observed in Waipuna Cave does not satisfactorily fit into this classification system. The physical arrangement of the drips belonging to the Organ Loft curtain (WP 1-1, WP 1-2, WP 1-3, WP 1-4, WP 1A, and WP 1B) and their lag response to rainfall (between 11 and 19 days) strongly suggest that these sites are hydrologically connected to each

other. Given that the curtain is part of the continuum from ceiling to floor (ca. 6 m height), it is likely that all its drip sites are mainly fed via diffuse flow through the limestone matrix, being a function of the primary porosity of the karst (Bradley et al., 2010). In contrast, drip sites WP-2, and WP-4, which are located in the upper gallery of the Organ Loft with a greater height of the ceiling, have a faster response (4 to 6 days) to antecedent rainfall, confirming that these drips are controlled by mainly by fracture flow. This is consistent with the clear identification of zones of structural weakness along the ceiling (physically

representing fault or joint-like structures) and the shorter vadose flow path at these locations. Given its location in the ceiling of the upper gallery and its intermediate response rate to rainfall of 11 days, WP-3 is most likely fed by combined fracture and matrix flow (Mahmud et al., 2018). The cross-correlation of the antecedent rainfall and the drip rates agrees with the cluster output for all drip sites except drip site WP-4, which has a response time of 4 days but clusters with the group of drip sites with a lag of 18 to 24 days. This can be explained by the limited size of the dataset: the drip rates at drip site WP-4 were

measured only manually, thus severely limiting the input for the cluster analysis compared to drip sites monitored with loggers. Our results indicate that the monitored drip sites respond to three different infiltration pathways via diffuse, fracture, and combined flow. Although the response time varies from days to 2–3 weeks, all drip sites forming stalagmites and feeding the flowstone reflect precipitation dynamics at sub-annual scale.

### 5.2 Rainwater isotope geochemistry

The distribution of the rainwater oxygen and hydrogen isotopes along the LMWL (Fig. 7) does not reveal a clear seasonal pattern. However, when comparing rainfall $\delta^{18}O$ values with the amount of precipitation across the entire monitoring period (Fig. 12, black line), we observe a positive relationship ($R^2 = 0.56$, $p = <0.0001$). The strongest correlations between rainfall amount and $\delta^{18}O$ values are observed in austral spring and summer ($R^2 = 0.68$ $p = 0.0002$, and $R^2 = 0.89$ $p = 0.0001$, respectively) when temperature is highest in the Waikato area (Fig. 12, green and orange lines). This amount effect is induced

by isotope fractionation due to partial re-evaporation of droplets falling through relatively dry air below cloud base in spring and summer (Dansgaard, 1964; Risi et al., 2008, Breitenbach et al., 2010). The amount effect varies over the year, with lower





re-evaporation and thus minimized fractionation during austral winter (this is reflected in lower R-values in from April to September, Fig. 12). These observations suggest that regional atmospheric conditions, associated with ENSO dynamics or strength of the Westerlies, can impose their signature on the isotopic composition of precipitation.

Dripwater $\delta^{18}O$ and $\delta D$ values closely reflect the mean isotopic composition of the rainwater (Fig. 7). The isotope signatures of dripwaters from the Organ Loft curtain lack seasonal patterns as they are decoupled from recharge rates by a significant epikarst store (Fig. 8b and c). Instead, buffering in the epikarst reservoir controls the isotopic composition of the water feeding the Organ Loft curtain. As highlighted in Figure 7, $\delta^{18}O$ values vary minimally around a mean dripwater $\delta^{18}O$ value of -5.6 ‰, which is only slightly lower than the average rainwater value (-5.15 ‰, Keller 2014). This similarity indicates rapid mixing

of freshly infiltrating water with older water in the epikarst, as also found in earlier studies from the Waitomo district (Williams and Fowler, 2002) and elsewhere (Mattey et al., 2008; Tremaine et al., 2016; Breitenbach et al., 2019). The observed buffering of dripwater towards the mean rainwater $\delta^{18}O$ value suggests that speleothem $\delta^{18}O$ ratios can be expected to reflect (multi-annual to multi-decadal changes in rainfall isotope geochemical patterns. Furthermore, it may be possible that speleothems from Waipuna Cave record a long-term temperature signal that is un-biased with regard to seasonal infiltration changes. On

the other hand, this pattern might also indicate that Waipuna Cave speleothem isotope geochemistry is insensitive to sub-seasonal changes in rainfall $\delta^{18}O$ and $\delta D$ ratios. Flowstone $\delta^{18}O$ values originating from water from these drips are unlikely to reflect atmospheric dynamics related to seasonal or ENSO variability, and other proxies must be used instead to identify these dynamics.

Waipuna Cave dripwaters do not show significant variations in $^{17}O_{excess}$ over time and most results overlap within analytical

errors (Supplement S5). Recent investigations into triple oxygen isotopes in mid-latitude rainfall have reported seasonal oscillations in $^{17}O_{excess}$ that have been attributed to changes in relative humidity at the moisture source (i.e., where the water vapour originates), or to swings between different moisture sources with evaporation occurring under different environmental conditions (Affolter et al., 2015; Uechi and Uemura, 2019). Unlike d-excess, $^{17}O_{excess}$ in rainfall is apparently almost exclusively controlled by relative humidity at the boundary layer, with insignificant temperature effects (Luz and Barkan,

2010). Thus, if there are seasonal changes in the dominant moisture source and origin of storms for the Waikato area, these would be likely to affect local precipitation and this variability could be recorded in Waipuna Cave dripwater. However, no significant variations in $^{17}O_{excess}$ are found over the studied period (September 2017 to October 2018), suggesting that the isotope values of meteoric water are homogenized in the epikarst, and that Waipuna Cave dripwater $^{17}O_{excess}$ is insensitive to (sub-) seasonal changes. We suspect that, as with $\delta^{18}O$ ratios, the interannual response of cave dripwater might be controlled

by long-term variations in the $^{17}O_{excess}$ of rainfall and changes in the relative importance of ENSO and the southern Westerlies. However, given the narrow range of $^{17}O_{excess}$ in rainwater in the mid-latitudes (normally < 30 per meg, Luz and Barkan, 2010) and the relatively large errors of current analytical methods (i.e., ~ 8 per meg), longer (i.e. multi-decadal) dripwater monitoring would be needed to test this hypothesis.

The fracture-flow fed drip sites WP-2, WP-3, and WP-4 are more sensitive to variations in surface conditions and were likely

affected by a moderate drought in the summer of 2016-2017 (Fig. 8a). 2016 was the warmest year on record for New Zealand, with average annual temperatures 0.5 to 1.2°C above normal (NIWA annual climate summary 2016). Rainfall was 50-79 % below average in December 2016, causing anomalously low soil moisture levels (NIWA summer 2016-17 report). Hence the January 2017 $\delta^{18}O$ values of -5.08 and -5.12 ‰ (Fig. 8a) might have been affected by the reduced infiltration caused by the higher evapotranspiration relative to precipitation.

The rapid decrease in dripwater $\delta^{18}O$ at the fast drip sites in March and April 2017 could have been caused by aquifer recharge (Fig. 8a). The decrease coincides with a period of increased precipitation, which would have quickly infiltrate the relatively





dry soil and entered the aquifer. This is consistent with the short water residence time of 5 days, and the greater degree of fracture flow and vadose zone influence at these sites compares with the slower drip sites.

### 5.3 Dripwater major and trace elements

Detecting short-term (sub-seasonal to annual) hydrological changes related to environmental conditions above Waipuna Cave requires sensitive (and ideally quantitative) proxies. In the following, we review the parameters we have measured in terms of their sensitivity.

Negative effective precipitation ($P_{eff}$), either from reduced rainfall or enhanced PET, can enhance degassing of $CO_2$ from the epikarst zone, and thus prior calcite precipitation PCP in the epikarst (Fairchild et al., 2000). Another factor potentially controlling PCP is cave ventilation. Enhanced ventilation removes moisture and $CO_2$ from the cave environment, which can result in < 100 % relative humidity (RH) and/or near-atmospheric $CO_2$ concentrations in cave air (Gázquez et al., 2016). Low cave air RH values can lead to dripwater evaporation, while low cave air $pCO_2$ can enhance dripwater degassing and formation of speleothems at the cave ceiling. Both processes can affect X/Ca and stable isotope ratios (Fairchild et al., 2006a; Breitenbach et al., 2015). Normally, the processes in the epikarst and in the cave act in concert and cannot be disentangled. Here, we show 380 that detailed monitoring of X/Ca dynamics in drip water can give valuable insights into the relative importance of these two zones, namely the epikarst and cave itself, for PCP intensity.

The PCP predictor line represent the modeled evolution of the $Ca_{aq}$ concentration which precipitates calcite in equilibrium as $pCO_2$ decrease from soil to the cave (Fairchild et al., 2006b). The Mg/Ca ratios of Waipuna Cave dripwaters closely follow the PCP predictor line (Fig.9a) vector. On the task of identifying the PCP the Mg/Ca and Sr/Ca ratios are plotted in Fig.9b and 385 a strong positive correlation is observed in Waipuna Cave dripwaters ($R^2 = 0.82$, $p = <0.001$) Sinclair et al., 2012; Tremaine and Froelich, 2013. This effect has also been widely identified in cave systems in Australia, with a climate similar to the Waitomo region. For example, Harrie Wood Cave dripwater Mg/Ca and Sr/Ca ratios show enhanced PCP during dry periods associated with El Niño, and reduced PCP during La Niña events (Tadros et al. 2016).

The degree of PCP could be expected to be linked to infiltration rates, with fracture flow being prone to more PCP because it 390 empties faster compared to seepage flow. As long as the epikarst remains water-filled, PCP would be minimized, whereas fast drying of the epikarst results in intrusion of soil air which might induce PCP. The fracture flow-fed drips can be distinguished from seepage flow-fed ones by lower Ca concentrations and increased scatter around the predicted PCP line (Fig. 9a). This can be explained by somewhat shorter interaction between the infiltrating water and the host rock. When comparing the elemental composition of the different drip sites (Fig. 9b), we observe that all drips show comparable Mg/Ca and Sr/Ca ratios, 395 suggesting that all drips are similarly affected by PCP. The different infiltration lag time of the individual drips thus does not appear to affect the extent of PCP in Waipuna Cave.

Although rainfall is evenly distributed throughout the year, a strong seasonal PCP signal is found in the dripwater for all drip sites across the whole monitoring period (Fig. 10). Lower Mg/Ca and Sr/Ca ratios occurred in the wettest months, when precipitation exceeded evapotranspiration. Conversely, higher Mg/Ca and Sr/Ca ratios are found in the driest months (i.e. 400 November to March) when the potential evapotranspiration exceeds rainfall and effective infiltration is negative (Fig. 10 and Supplement S7). Hydrological changes thus govern epikarst PCP, which in turn controls dripwater Mg/Ca and Sr/Ca ratios (Fig. 9b). This observation supports our hypothesis that Waipuna Cave dripwaters are capable of registering changes in local hydrology, with seasonal differences being most strongly expressed in Sr/Ca ratios. Changes in Sr/Ca ratio potentially reflect the interplay of PCP and enhanced selective Sr leaching (incongruent dissolution), which both operate to increase Sr/Ca in the 405 drier months (Sinclair et al., 2012), while the wetter months are characterised by infiltration and reduced selective Sr leaching (congruent dissolution) (see Section 5.5).





### 5.4 Cave ventilation

Monitoring of temperature and $CO_2$ between June 2017 and June 2018 shows that Waipuna Cave ventilation is driven by

changes in the density of internal and external air in response to seasonal external temperature (Fig. 11), i.e., Waipuna Cave is a barometric cave (Fernández-Cortés et al., 2008). During late spring and summer (November 2017 to May 2018), cave air is colder than surface air ($\Delta T < 0$) (Fig. 11c). A greater relative density of the cave air and the pressure difference compared to the surface air creates a cold air 'lake' within the cave. This cold air mass is isolated from the warmer, less dense, exterior air (i.e., isolation period) due to the geometry of the cave (Fig. 13). The cold, stagnant cave air inhibits exhalation of $CO_2$ released

from the dripwater, which then accumulates in the cave atmosphere. Inversely, from autumn to early spring (June 2017 to October 2017), a positive $\Delta T$ (i.e. warmer cave air relative to the surface, though still colder than summer cave air) leads to barometric ventilation of cave air (Fig. 11c). Due to the pressure gradient, cool and dense surface air will sink into the cave, whilst rising warm cave air leaves the cave. The intensified air exchange promotes $CO_2$ extraction from the cave. This effect is reflected in the positive relation between $T_{cave\ air}$ and $CO_2$ values ($R^2 = 0.67$, $p = 0.04$). These two phases of cave ventilation

dynamics fit the chimney circulation model (Fairchild and Baker, 2012) and have been observed in similar climatic settings in the USA (Oster et al., 2012), India (Breitenbach et al., 2015) and Spain (Gazquez et al., 2017) among others.

Furthermore, we find that during the period with negative $P_{eff}$, normally the summer season, the relationship between Mg/Ca and Sr/Ca ratios is more pronounced, reflected in a higher $R^2$ value ($R^2 = 0.92$, $p < 0.001$) and a steeper slope compared to the winter season ($R^2 = 0.52$, $p < 0.001$), when this relationship is less strong, and the slope is lower (Supplement S8). Together

with lower dripwater X/Ca values in the winter season, this suggests a less significant role for PCP at times of higher ventilation and $CO_2$ changes in Waipuna Cave. Since all Mg/Ca and Sr/Ca ratios fall along the PCP line during the months of reduced ventilation (November–March), it seems that enhanced cave ventilation does not affect PCP.

### 5.5 A whisper of La Niña in Waipuna Cave

We have demonstrated that in Waipuna Cave, Mg/Ca and Sr/Ca ratios are sensitive PCP indicators. Here we discuss how they

potentially react to infiltration changes governed by ENSO dynamics.

A plot of Mg/Ca versus Sr/Ca ratios displays two clusters, each along a clear trend (Fig. 9b and 14). Blue-coloured symbols represent samples collected between late February 2018 and the end of August 2018, a period with above-average rainfall, likely related to a La Niña event that developed in December 2017. These conditions prevailed over the following months (January to March 2018) (NIWA 2018a). Even though La Niña dissipated in March 2018, it still affected early autumn

circulation patterns in the central North Island, expressed generally by stronger than usual northeasterly winds, above-normal temperatures (mean > 1.2°C), well above normal rainfall (> 149 % of normal), and much higher soil moisture levels for this time of year (NIWA, 2018b). Dripwater samples collected before February 2018 and after August 2018 (ENSO-neutral conditions), plot on the main PCP trend, but samples collected during the between February 2018 and August 2018 (during La Niña event decay) plot on a distinct line (Fig. 14).

It therefore seems possible to identify ENSO events by singling out different regression trends in Sr/Ca-Mg/Ca space (Fig. 14). Our data, combined with meteorological information, suggest different behavior of Sr/Ca during the warm/wet La Niña event. A comparison of intercept values of the two trendlines suggests that wet La Niña conditions promoted higher effective infiltration, thereby reducing Sr availability.

Mg/Ca and Sr/Ca ratios are lower during winter, the wettest months with the lowest PET. Conversely, Sr/Ca and Mg/Ca ratios

are higher in the higher PET, drier summer months (Fig. 10 and Supplement S8). We postulate that in Waipuna Cave, the La Niña climate mode, although short-lived, has a strong influence on Sr/Ca variation that produces an overprint on PCP dynamics. In the case of the 2017 La Niña event, we interpret the data to indicate that the extra infiltration associated with the





event fundamentally altered the regime of host rock dissolution, thereby decreasing Sr availability (Fairchild and Treble 2009) in a manner consistent with congruent host rock dissolution and reduced selectivity in Sr leaching (Fairchild et al., 2000). In summary, we argue that hydrological change associated with ENSO, which amplifies the length of the 'wet' time window, should modulate Sr/Ca to a greater extent than seasonal changes.

## 6 Conclusions

The results of a three-year long multi-parameter monitoring campaign in Waipuna Cave help to characterise the sensitivity of the cave with respect to external climatic changes occurring on intra-annual time scales in the North Island of New Zealand. The monitored parameters include drip rates, cave air temperature, dripwater trace elements, water stable isotopes, and cave air $p$CO$_2$. These were compared to meteorological data from nearby stations. Based on geochemical and drip rate data, we identify three distinct infiltration pathways, diffuse flow, fracture flow, and combined flow, with lagged response to antecedent rainfall of 18–15 days, 4–6 days, and 11 days, respectively. Waipuna Cave thus quickly reacts (within 24 days) to external precipitation variability and is sensitive to sub-seasonal changes in epikarst hydrology.

Dripwater isotope composition in Waipuna Cave reflects the mean rainwater δ$^{18}$O and δD values. Mixing processes in soil and epikarst obscure any seasonal isotopic signal in the dripwater. However, long term (i.e., inter-annual to decadal) atmospheric changes are very likely recorded by speleothem calcite δ$^{18}$O ratios in Waipuna Cave. Because local spring and summer rainfall isotopes values are influenced by the amount effect, pronounced droughts can affect the isotopic composition of the dripwater, and that signal may be recorded in speleothems.

Dripwater Mg/Ca and Sr/Ca ratios are modulated by PCP and reflect local hydrological changes. Higher Mg/Ca and Sr/Ca ratios reflect periods of reduced effective infiltration from November to March when the potential evapotranspiration exceeds local rainfall amount. The relationship between Mg/Ca and Sr/Ca ratios may be affected by ENSO variability, with wetter conditions and reduced PCP occurring during La Niña events reflected in lower Mg/Ca-Sr/Ca slopes. This relationship may thus be a sensitive geochemical tracer of ENSO dynamics. However, longer monitoring is required to validate this interpretation.

Surface air temperature changes govern cave ventilation in Waipuna Cave. Enhanced ventilation occurs between April and October (austral winter) when the surface air temperature is lower than in the cave. During austral summer surface air temperatures are higher than cave air temperatures, resulting in reduced ventilation by virtue of a cold cave air lake. The Waipuna Cave ventilation pattern is an important factor controlling dripwater degassing, cave air CO$_2$ dynamics, speleothem growth rates, and isotope fractionation.

The findings of this study on the hydrochemistry in Waipuna Cave establish a baseline that will allow interpretation of speleothem-based proxy records at seasonal and interannual scales to reconstruct local hydrological changes as well as regional dynamics e.g. ENSO events. Longer-term monitoring is required in order to better constrain the effects of synoptic-scale environmental fluctuations on speleothem records from Waipuna Cave and nearby caves.

### Data and samples availability

Data reported here will be made fully available to the public via a public domain platform. Additionally, data can be requested from the corresponding author (CN, cinthya.navafernandez@rub.de).

### Author contributions



CN conducted fieldwork, collected the samples and data, analysed the data, and prepared the original manuscript. AH designed and carried out the cave monitoring programme, conducted fieldwork, and supervised the study. FG conducted the stable water analyses, and contributed to the discussion of the results. OK participated in the fieldwork, contributed to the discussion, helped with figures, supervised the study, and contributed to the discussion, NM helped with statistical analysis and discussion. BF
conducted fieldwork, helped with the statistical analysis and writing. JH contributed with fieldwork and structure from motion images. AP and BW helped in the cave monitoring effort. AF carried out the major and trace elements analysis and contributed to the discussion. DH provided laboratory resources and helped acquisition of funding. AI contributed with the editing process. SB designed the monitoring programme, supervised the study, collected samples, and contributed to the interpretation, visualization, and preparation of the manuscript.

**Competing interests**

The authors declare that they have no conflict of interest.

**Acknowledgments**

Thanks to Ingrid Lindeman, Inken Heidke, Jackson White for their valuable fieldwork contributions. We thank Peter and Libby Chandler for their permission to access cave and their ongoing support of research. C.N.F. acknowledges financial support
from the German Academic Exchange Service (DAAD). F.G. was financially supported by the "HIPATIA" research program of the University of Almería. This study received funding from the European Union's Horizon 2020 Research and Innovation programme under the Marie Skłodowska-Curie grant agreement No 691037 and Royal Society of New Zealand grant agreement RIS-UOW1501, and Rutherford Discovery Fellowship (RDF-UOW1601) to A.H.

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



**Table 1.** Summary of drip site morphological characteristics, location in the cave, response time to antecedent rainfall (values in bold were calculated from the logger records, the rest from manual drip counts), coefficient of variation % (CV), and hydrological behavior.

| Drip Site | Description | Location | Height to the ceiling (m) | Mean discharge ($\mu Ls^{-1}$) | Response time to rainfall (days) | CV (%) | Flow pattern |
|---|---|---|---|---|---|---|---|
| WP 1-1 | Part of the flowstone curtain | Organ loft | 6 | 63.35 | **17** | 52.9 | Diffuse flow |
| WP 1-2 | Part of the flowstone curtain | Organ loft | 6 | 43.74 | **15** | 85.1 | Diffuse flow |
| WP 1-3 | Part of the flowstone curtain | Organ loft | 6 | 105.57 | **16** | 56.2 | Diffuse flow |
| WP 1-4 | Part of the flowstone curtain | Organ loft | 6 | 22.13 | 18 | 53.9 | Diffuse flow |
| WP 1A | Part of the flowstone curtain | Organ loft | 6 | 83.99 | 18 | 36.3 | Diffuse flow |
| WP 1B | Part of the flowstone curtain | Organ loft | 6 | 51.83 | 11 | 48.2 | Combined flow |
| WP FB | Flowstone bottom | Organ loft | 8 | 30.08 | 9 | 81.8 | Diffuse flow |
| WP-2 | Independent stalactite | Upper gallery | 9 | 59.57 | **4** | 149.4 | Fracture flow |
| WP-3 | Independent stalactite | Upper gallery | 9 | 91.62 | 11 | 31.3 | Combined flow |
| WP-4 | Independent stalactite | Upper gallery | 12 | 32.07 | 6 | 53.2 | Fracture flow |





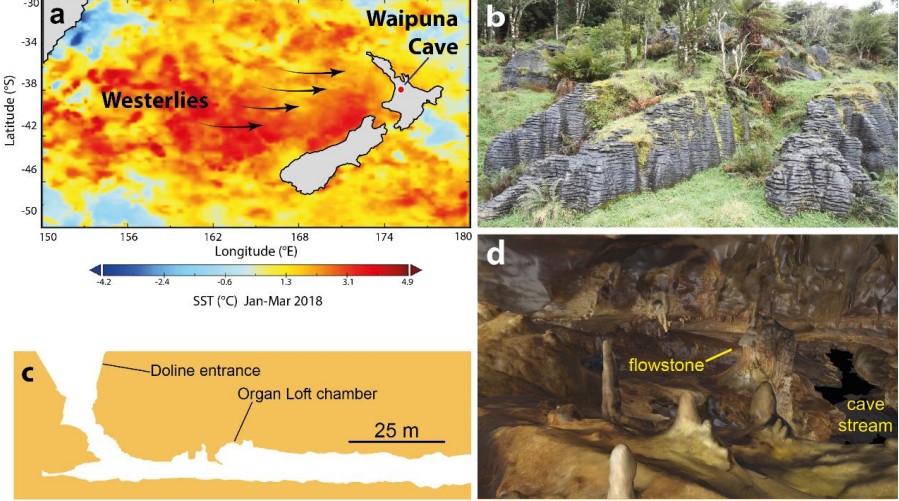


**Figure 1. a)** geographical location of Waipuna Cave and its climatic settings during La Niña conditions in summer 2018, with the direction of the Westerlies. Source data: OSTIA. **b)** pancake limestone outcrop landscape above Waipuna Cave. **c)** cross section through the entrance and first stretch of Waipuna Cave. **d)** 3D model of Organ Loft Chamber photo taken by John Hellstrom using structure from motion (SfM) mapping.


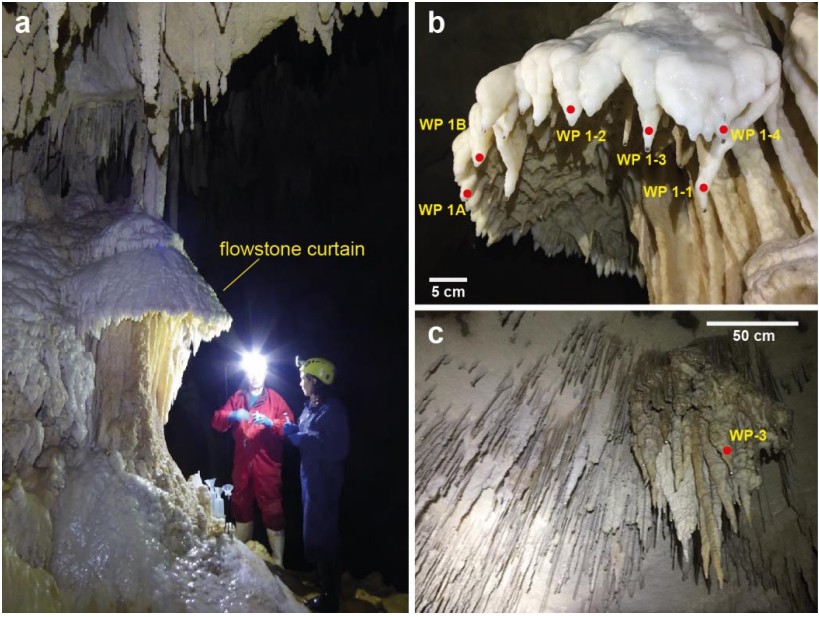

**Figure 2. a)** flowstone formation in the "Organ Loft" chamber (Photograph courtesy of I. Heidke). **b)** drip sites at the flowstone curtain. **c)** stalactite cluster where drip site WP-3 is located, hanging from the ceiling 6 m high in an inclined plane.






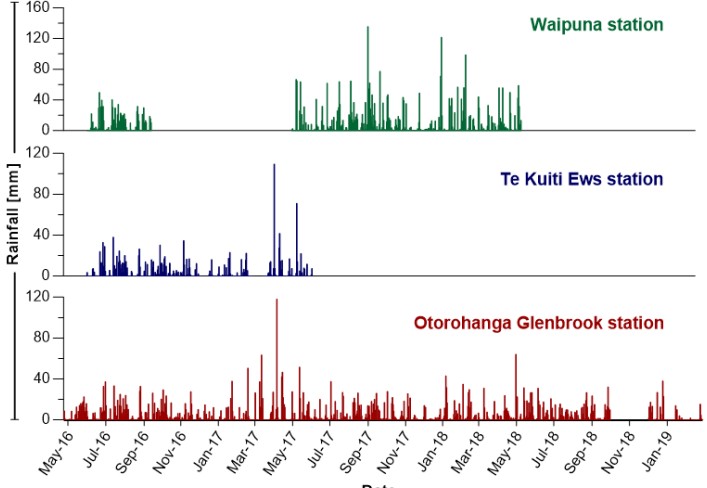

**Figure 3. Daily precipitation from the Waipuna meteorological station (no data is available for October and November 2018 due to instrument failure), Otorohanga Glenbrook and Te Kuiti Ews stations (data from NIWA National Climate Database, www.clifo.niwa.co.nz).**

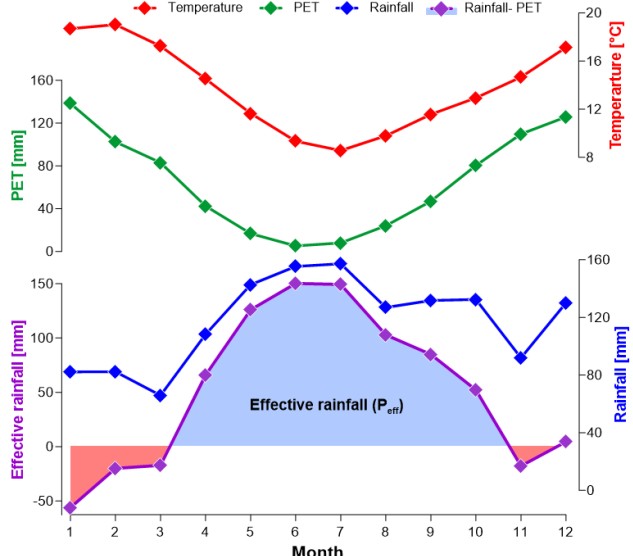

**Figure 4. Mean monthly temperature (red line), precipitation (blue line), and Priestley-Taylor potential evapotranspiration (PET) recorded at Te Kuiti Ews station between 2002 and 2019 (green line) (http://cliflo.niwa.co.nz), and difference between rainfall and PET (purple line). The shaded blue area represents the effective rainfall ($P_{eff}$).**


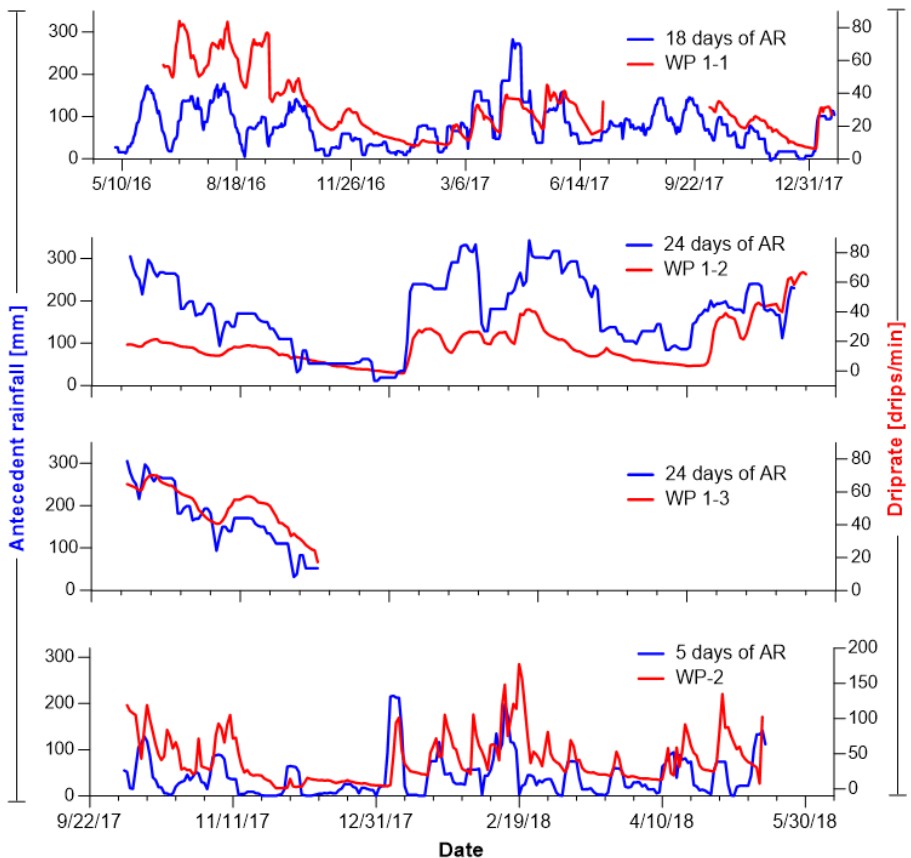

**Figure 5. Drip rates recorded by the loggers (red line) for the drip sites WP 1-1, WP 1-2, WP 1-3 and WP-2, plotted with their best-fit of antecedent effective-rainfall days (blue).**


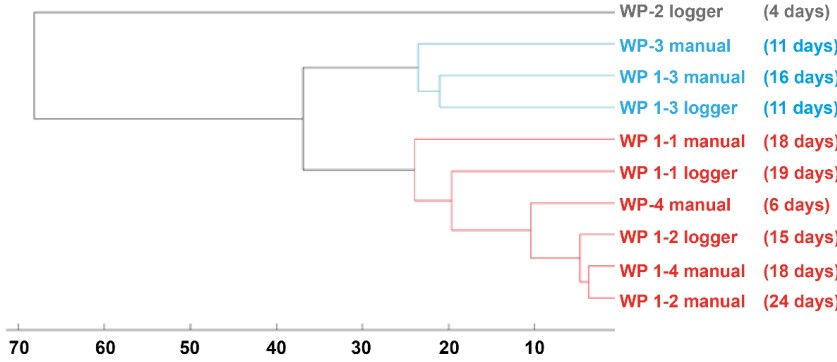

**Figure 6. Results of cluster analysis of the drip site discharge time series, indicating the number of lag days calculated from the cross-correlation analysis. The gray cluster shows the drip site with the shortest lag to the antecedent rainfall (4 days), the blue cluster groups the drip sites with between 11 and 16 days lag to antecedent rainfall, and the red cluster includes drip sites with a lag to antecedent rainfall between 15 and 24 days, with the exception of WP-4 (see main text).**




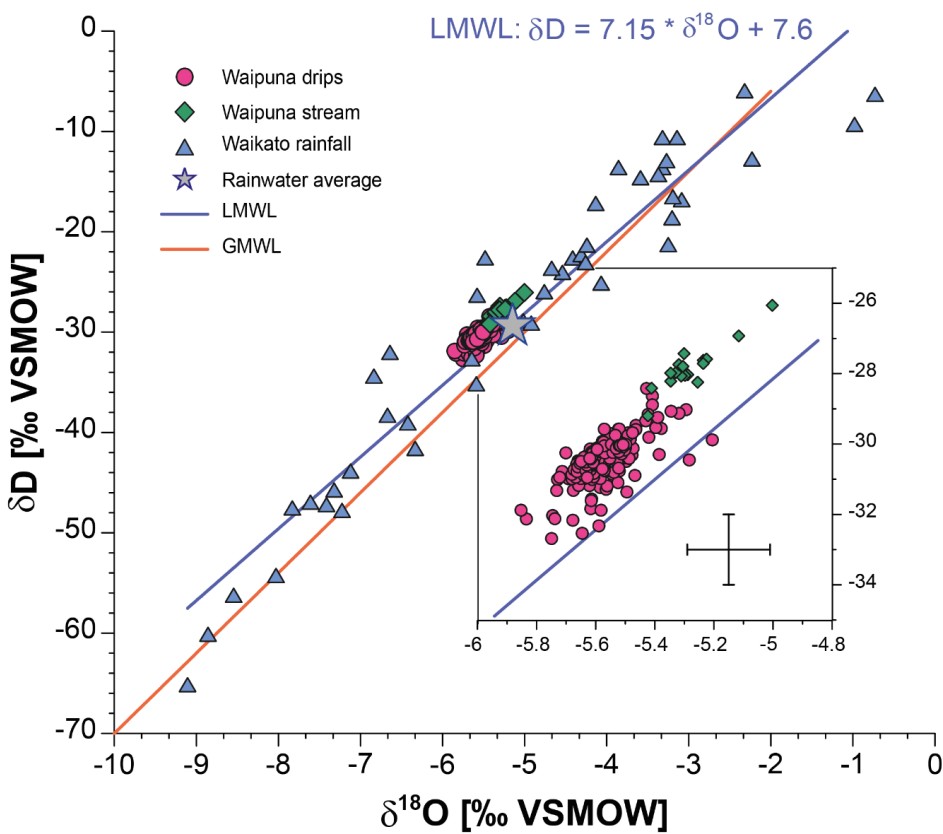


**Figure 7. Cross-plot of δD versus δ¹⁸O in dripwater (pink circles), the Waikato region precipitation (blue triangles) and Waipuna stream (green diamonds). All cave waters fall in a very narrow range (inset) and within error on the Local Meteoric Water Line (blue line, Keller et al., 2014). The cross in the inset shows the 2σ uncertainties for δ¹⁸O and δD.**



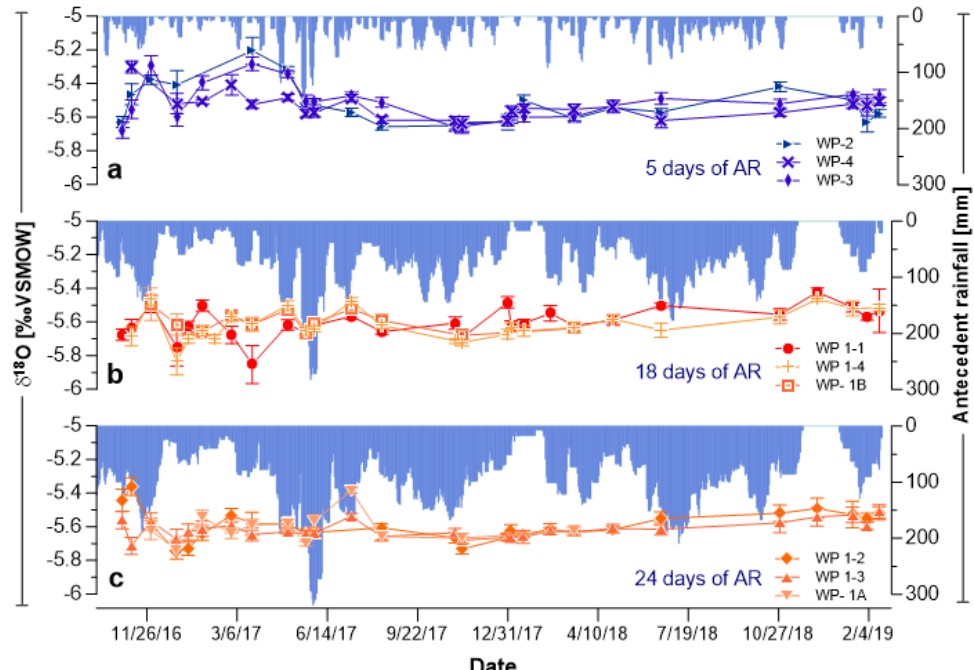

Figure 8. Dripwater $\delta^{18}O$ time series of all drips grouped according to the three main response lags (5, 18, and 24 days) to antecedent rainfall (AR) at Otorohanga Glenbrook station (blue vertical bars). a) drip sites WP-2, WP-3, and WP-4 (5 days' antecedent rainfall). b) drip sites from the flowstone curtain, WP 1-1, WP 1-4 and WP 1B (18 days' antecedent rainfall). c) drip sites from the flowstone curtain WP 1-2, WP 1-3 and WP1A (24 days' antecedent rainfall). The Otorohanga rainfall record covers the period between September 2016 and January 2019; no data are available for October and November 2018.

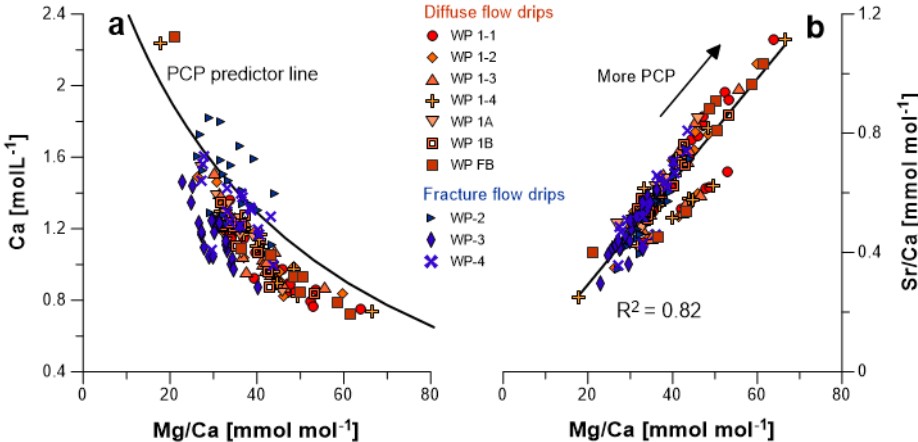

Figure 9. Waipuna Cave dripwaters sampled during the monitoring period October 2016 to January 2019. a) Ca concentration versus Mg/Ca ratios. b) Mg/Ca versus Sr/Ca, $R^2 = 0.82$, $p < 0.001$. The orange symbols correspond to diffuse flow drip sites and blue symbols signify fracture flow drip sites. The two different trends evident in the relationship between Mg/Ca and Sr/Ca will be discussed in section 5.5.



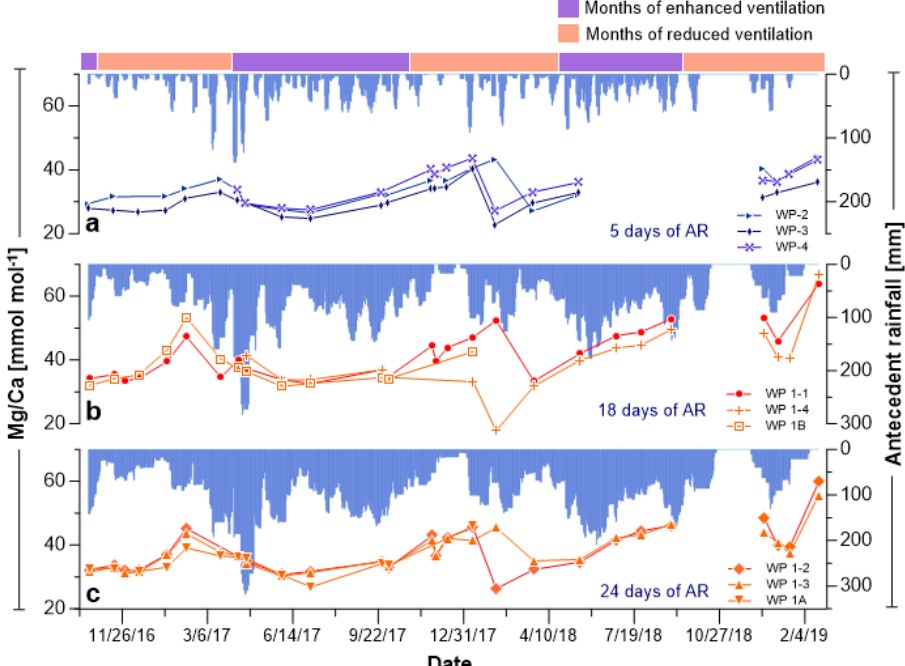

**Figure 10. Dripwater Mg/Ca ratios time series of all drips grouped according to the three main response lags (5, 18, and 24 days) to antecedent rainfall (AR) at Otorohanga Glenbrook station (blue vertical bars). a) drip sites WP-2, WP-3, and WP-4 (5 days' antecedent rainfall). b) drip sites from the flowstone curtain WP 1-1, WP 1-4 and WP 1B (18 days' antecedent rainfall). c) drip sites from the flowstone curtain WP 1-2, WP 1-3 and WP1A (24 days' antecedent rainfall). The Otorohanga rainfall record covers the period between September 2016 and January 2019; no data are available for October and November 2018.**



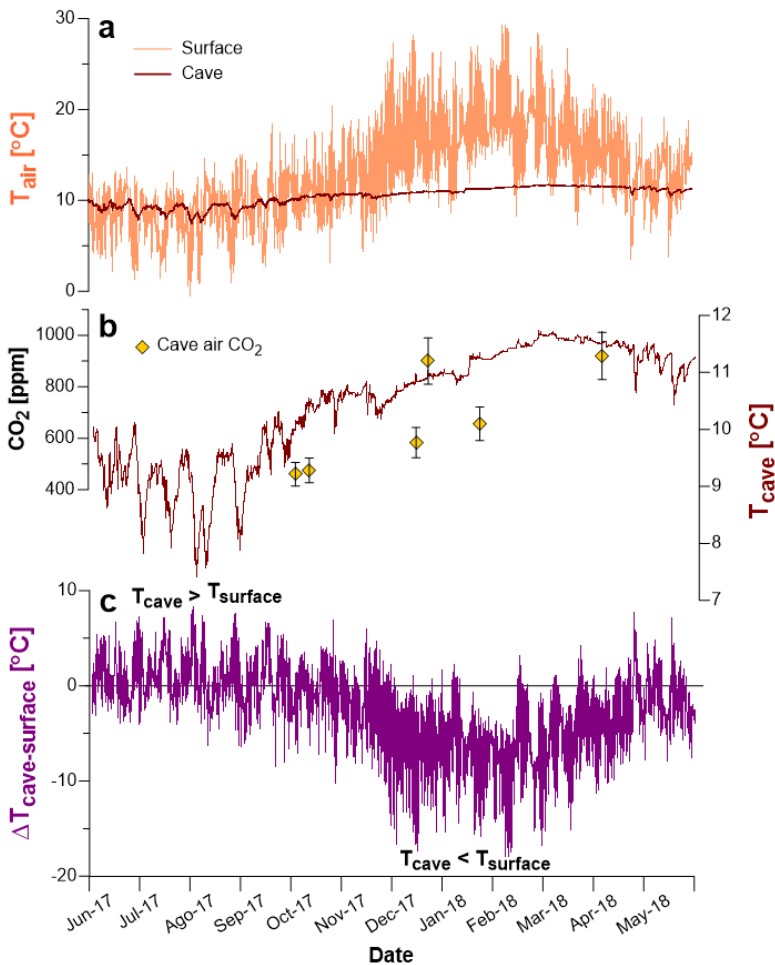

**Figure 11. Comparison of meteorological parameters. a) daily mean air temperature at Waipuna station and in Waipuna Cave (in Organ Loft chamber). b) manual cave air CO₂ concentrations and daily mean air temperature in the Organ Loft chamber. c) calculated temperature difference ΔT between cave air and surface air from June to May 2018. During winter, the cave is warmer, and during summer, the cave is colder compared to the surface air.**

730





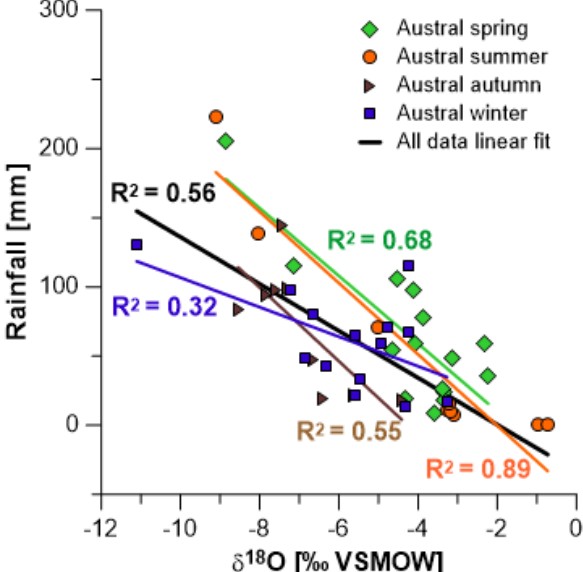

**Figure 12. Cross-plot of precipitation amount versus $\delta^{18}O$ in rainfall for the period August 2007 to December 2009 from the Waikato region (Data source: Keller, 2014). The black line indicates the correlation if all data are considered, while the colored symbols and regressions relate to the different seasons.**

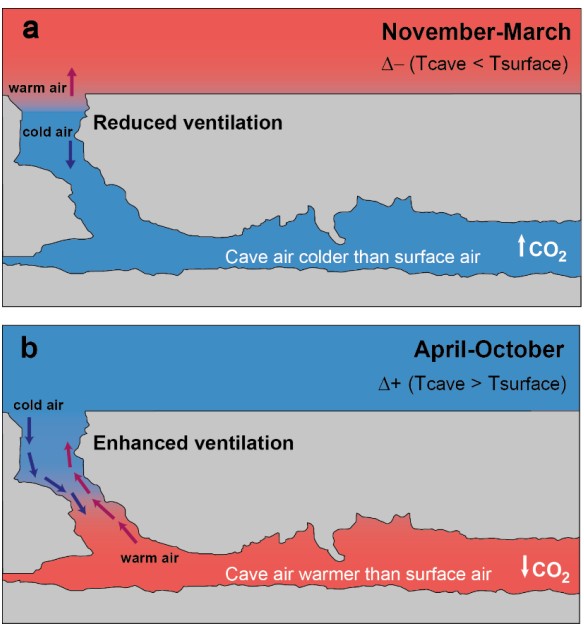

**Figure 13. Conceptual model of Waipuna Cave ventilation and cave air $CO_2$ dynamics. The cave air is warmer in summer colder in winter. However, in summer cave air is colder than surface air ($\Delta+$) and in winter cave air is warmer than surface air ($\Delta-$). a) spring and summer when warmer surface conditions compared to the cave interior leads to a stagnant cold air lake and maximum $CO_2$ values in the cave, and b) autumn and winter conditions characterised by low surface and relatively higher cave air temperatures, facilitating barometric ventilation and exhalation of cave air $CO_2$ to the surface.**





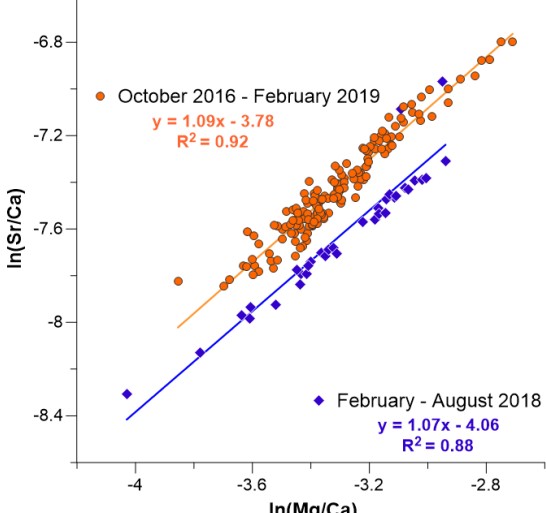

745 **Figure 14. Dripwater ln(Mg/Ca) versus ln(Sr/Ca) ratios in Waipuna Cave. Blue diamonds indicate samples collected between late February 2018 to the end of August 2018, orange circles all other samples collected between October 2016 and February 2019.**