# Peer review of "Pacific climate reflected in Waipuna Cave dripwater hydrochemistry"

_Hydrology and Earth System Sciences, 2019_

## Referee Comment (RC1) · Anonymous Referee #1 · 27 Jan 2020

This manuscript reports the results and interpretation of a multi-year cave monitoring study in an ENSO-sensitive region of New Zealand. With a few minor exceptions, largely relating to oddly used commas, the writing is clear and the paper is well organized. The figures are appropriate and nicely constructed. I list specific comments below, none of which should require much work to incorporate. As I mention a couple of times, it seems odd that barometric pressure data were not included. The authors reference a 2008 study of the same cave and I argue strongly that data from this study should be included here, even if they don't overlap with the years that $CO_2$ and dripwater were measured here.

With respect to the particular criteria of the journal:

1. Does the paper address relevant scientific questions within the scope of HESS?

[Figure]

YES 2. Does the paper present novel concepts, ideas, tools, or data? A BIT 3. Are substantial conclusions reached? A BIT 4. Are the scientific methods and assumptions valid and clearly outlined? LARGELY 5. Are the results sufficient to support the interpretations and conclusions? LARGELY 6. Is the description of experiments and calculations sufficiently complete and precise to allow their reproduction by fellow scientists (traceability of results)? YES 7. Do the authors give proper credit to related work and clearly indicate their own new/original contribution? YES 8. Does the title clearly reflect the contents of the paper? YES 9. Does the abstract provide a concise and complete summary? YES 10. Is the overall presentation well structured and clear? YES 11. Is the language fluent and precise? YES 12. Are mathematical formulae, symbols, abbreviations, and units correctly defined and used? YES 13. Should any parts of the paper (text, formulae, figures, tables) be clarified, reduced, combined, or eliminated? A BIT 14. Are the number and quality of references appropriate? YES 15. Is the amount and quality of supplementary material appropriate? YES

Specific Comments Line 29 – and $CO_2$ concentration

– diffuse flow, fracture flow, and combined flow

– is buffering the right word? Perhaps homogenization?

– atmospheric-oceanic

– does eastern NZ refer only to the North Island? If not, then leave as is, but given that a lot of paleoclimate work (pollen, glaciers, speleothems, etc) has been done on the South Island, it is important to distinguish between N Island-only signals and those that impact all of NZ.

– short time span (beginning early 1800s).

– priority, both because

– I would add that we still don't really understand the nature of ENSO over the last few millennia and ENSO-sensitive sites capable of providing meaningful reconstructions are highly valued.

– either write atmospheric-oceanic or atmosphere-ocean

– I don't understand this claim; can't one calibrate d18O in snow atop glaciers v temp? Coral geochemistry vs SST? Marine core top calibration is commonly done. Tree rings seem to be one of the few records truly complicated by modern calibration owing to (likely) CO2 fertilization effects.

– hydrology, and hydrochemistry is critical

– in speleothems because numerous studies have shown imperfect replication between coeval stalagmites or plate-grown calcite, as well as differences in dripwater chemistry.

– in my opinion, speleothem paleoclimate work is shifting toward an understanding that cave hydrology/dripwater geochemistry is a prerequisite for meaningful interpretation of speleothems. However, it is not enough. Speleothem records must also be replicated. I would like to see some mention of that here.

– please differentiate more fully between fracture flow and "iii) conduits with high flow rates"

– increasing PCP depending on their partition coefficients in calcium carbonate

– here you write "south-western", but earlier you use SW. Be consistent, but don't hyphenate southwestern.

– not sure Borneo is the southwestern Pacific. It's equatorial to slightly NH.

– need a verb after iii)

– are you sure "typic orthic allophanic" is the appropriate way to describe these soils?

143.- and recorded

– 22 km is a pretty long distance. Why include this station? Why not just the one 13 km from the cave?

– please expand on the methods for monthly rainwater collection? What was used to minimize evaporation?

– was counted

– of variation (CV)

– of dripwater and stream water

– CV has already been defined

– I understand that the data are significant to three places, but it is distracting to include values in the hundredths place for delta values. It is easier to remember and no less significant to the story if you write "varied between -3.0 and -2.4‰$\ldots -5.9 and -5.2 for d18O \ldots$"

258-261 – why not include mean and standard deviation here? Much more informative than range alone.

– PCP has already been defined

– see my earlier comments regarding reporting the hundredths (or thousandth!) place

– this correlation deserves at least a little bit of explanation. Would have been nice if the cave monitoring had included barometric pressure. . .

– this is a long, but incomplete, sentence. "This work is aimed"

– physiochemical?

– I am not convinced the data "confirm" anything, but they do "suggest" or "argue for" fracture flow

– this discussion of amount effects in the rain data is too abbreviated. The origins of amount effects have been demonstrated to reflect any of a suite of drivers, including storm track, and these should be fleshed out here in more detail.

– why the open paren?

– again, it would have made a great deal of sense to have installed a barometric pressure logger within and outside the cave to address questions of ventilation.

– put references in parentheses

– Wood Cave, located # km (direction) from Waipuna Cave,

– Replot some of the data from Fernandez-Cortes et al., 2008 in this paper to illustrate the effects of air pressure.

– Is this section title, while poetic, a touch too flowery?

Figure 3 – please add to the figure itself the intervals for which no data are available. Don't rely solely on mentioning this in the caption.

---

## Author Comment (AC1) · 3 Mar 2020

Dear Referee,

Thank you for the very helpful comments which we consider in our revised manuscript. Please find below our point-by-point response to all your comments and those of reviewer 1. We submit a substantially revised manuscript following these comments, which indeed helped to improve our paper. We believe that this contribution will be well-regarded by readers of HESS, and hope you find it suitable for publication.

Thank you for your time and consideration,

Sincerely, Cinthya Nava Fernandez, on behalf of all co-authors.

Anonymous Referee #1 Received and published: 27 January 2020 This manuscript reports the results and interpretation of a multi-year cave monitoring study in an ENSO-sensitive region of New Zealand. With a few minor exceptions, largely relating to oddly used commas, the writing is clear and the paper is well organized. The figures are appropriate and nicely constructed. I list specific comments below, none of which should require much work to incorporate. As I mention a couple of times, it seems odd that barometric pressure data were not included. The authors reference a 2008 study of the same cave and I argue strongly that data from this study should be included here, even if they don't overlap with the years that CO2 and dripwater were measured here.

Specific Comments Line

- 29 and CO2 concentration Done
- $35-\mbox{diffuse}$  flow, fracture flow, and combined flow Done

– is buffering the right word? Perhaps homogenization? Done, we rephrased the sentence

– atmospheric-oceanic Done

– does eastern NZ refer only to the North Island? If not, then leave as is, but given that a lot of paleoclimate work (pollen, glaciers, speleothems, etc) has been done on the South Island, it is important to distinguish between N Island-only signals and those that impact all of NZ. We agree in highlighting the importance to distinguish between both islands. In line 55 'NZ' refers to both islands, but for the next sentence the effects of La Niña conditions are specifically related to the North Island. We specify this more clearly now in the text.

- 61 short time span (beginning early 1800s) Done
- 61 priority, both because Done
– I would add that we still don't really understand the nature of ENSO over the last few millennia and ENSO-sensitive sites capable of providing meaningful reconstructions are highly valued. We agree and added a short hint in the main manuscript (red text): Since the nature of ENSO over the last few millennia remains poorly understood, ENSO-sensitive study sites that provide long, robustly datable proxy reconstructions are urgently needed.

- either write atmospheric-oceanic or atmosphere-ocean Done

– I don't understand this claim; can't one calibrate d18O in snow atop glaciers v temp? Coral geochemistry vs SST? Marine core top calibration is commonly done. Tree rings seem to be one of the few records truly complicated by modern calibration owing to (likely) CO2 fertilization effects.# To explain this more clearly we adjusted the main text as follows: Speleothems provide reliable continental palaeoclimate records because they allow for modern calibrations linking palaeo-data from stalagmites with meteorological and direct in-cave monitoring, thus making it possible to trace climatic signals from the surface to the speleothem at timescales from seasonal (Frappier et al., 2002) to orbital (Meckler et al., 2012; Mattey et al., 2008).

- hydrology, and hydrochemistry is critical Done

– in speleothems because numerous studies have shown imperfect replication between coeval stalagmites or plate-grown calcite, as well as differences in dripwater chemistry. d. Done, we added this: Monitoring of modern cave environments, encompassing ventilation, hydrology and hydrochemistry is critical for reliable interpretations of palaeo-environmental proxies preserved in speleothems because numerous studies have shown imperfect replication between coeval stalagmites, as well as differences in dripwater composition (McDermott, 2004; Fairchild et al., 2006a; Breitenbach et al., 2015).

– in my opinion, speleothem paleoclimate work is shifting toward an understanding that cave hydrology/dripwater geochemistry is a prerequisite for meaningful interpre-

HESSD
tation of speleothems. However, it is not enough. Speleothem records must also be replicated. I would like to see some mention of that here. We agree with the reviewer that replication is of great value, but we argue that a lack of agreement between two stalagmites from the same cave does not mean that this cave does not carry environmental and climatic information. It simply means that detailed monitoring is vital to understand the underlying physical processes that lead to these differences. We added a clarifying note in the main text: The characterization of infiltration pathways is an essential prerequisite for delineating the processes that can modulate dripwater chemistry, i.e. the degree of water-rock interaction taking place in the epikarst and the climate signal transferred by the dripwater to the speleothems. The climatic signal transferred to speleothems can vary even between speleothems from the same chamber, which emphasizes the need for detailed monitoring. As every stalagmite records the conditions that occur in the epikarst and are transported by the feeding dripwater, in-depth understanding of the forcing mechanisms is vital to understand the differences between non-replicating records.

– please differentiate more fully between fracture flow and "iii) conduits with high flow rates" We explain this in more detail now: The physical proprieties of the karst zone define the different levels of porosity which is primarily characterised by intergranular pore space, while secondary porosity is associated with joints and fractures, and tertiary porosity with solution-enhanced conduits (Ford and Williams, 2007). Seepage water experiences one or a combination of these different porosity levels, which determine hydrological pathways (Fairchild and Baker 2012).

– increasing PCP depending on their partition coefficients in calcium carbonate Done, we added a short explanation: Flow routing to speleothem drip points is the first-order control on dripwater hydrochemistry, with particular relevance for trace elements and other proxies of prior calcite precipitation (PCP) (Fairchild et al., 2000; Wassenburg et al., 2012). PCP serves as a proxy system for moisture availability (Magiera et al., 2019), and affects a range of trace elements, which may either become more
concentrated (increasing X/Ca) or diluted in solution (decreasing X/Ca) with increasing PCP depending on their partition coefficients between solution and calcium carbonate (Hartland and Zitoun, 2018).

– here you write "south-western", but earlier you use SW. Be consistent, but don't hyphenate southwestern Done

– not sure Borneo is the southwestern Pacific. It's equatorial to slightly NH This is correct and now adjusted in the text.

– need a verb after iii). Done, the new main text reads as follows: Our study has three consecutive objectives: i) characterisation of the dripwater chemistry, including major and trace elements (Mg/Ca and Sr/Ca) and isotope geochemistry (ĩĄď17O, ĩĄď18O, ĩĄďD and d-excess); ii) identification of the mechanisms controlling dripwater chemistry; and iii) understanding the relationship between dripwater chemistry and variations in precipitation, and seasonal and interannual (ENSO) climate conditions.

– are you sure "typic orthic allophanic" is the appropriate way to describe these soils? Yes, we follow the soil classification established by the TRC (Taranaki Regional Council) in the cover soil map for Waipuna Cave location. S-map Soil Report (Manaaki Whenua Landcare Research) https://smap.landcareresearch.co.nz

143.- and recorded Done

– 22 km is a pretty long distance. Why include this station? Why not just the one 13 km from the cave? We use Othorohanga station data because other datasets contain gaps. Othorohanga station is in a neighbouring area that shows similar rainfall patterns and this meteorological record has the most complete record of all variables considered in our work during the monitoring period.

– please expand on the methods for monthly rainwater collection? What was used to minimize evaporation? Thanks for spotting this. This sentence was not deleted by mistake since we have very few rain water measurements which we chose not to

HESSD
discuss in this paper and this information has been removed.

- 170 was counted Done
- 174 of variation (CV) Done
- 178 of dripwater and stream water Done
- 233 CV has already been defined Done

- I understand that the data are significant to three places, but it is distracting to include values in the hundredths place for delta values. It is easier to remember and no less significant to the story if you write "varied between -3.0 and -2.4‰ ...-5.9and-5.2ford18O...00258 - Weagree, and we changed this accordingly

– why not include mean and standard deviation here? Much more informative than range alone. We have added some more detail in the main text to allow for better evaluation, but we think that the range is the more informative parameter in this context, as it quantifies the total observed variation in the dripwater d18O signal over the sampling period. The standard deviation around the mean is here less helpful, as it condenses the information too much in the sense that individual observations are smoothened. Even a few observations in either direction contain helpful information, which is better reflected in the total range. The main text now reads: Between September 2016 and June 2017, drip sites WP 1-1, WP 1-2, WP 1-3, WP 1-4, WP 1A, and WP 1B show little variability in dripwater d18O and dD, ranging 0.5 % (-5.4 to -5.9 % in d18O and 3.9 ‰ (-28.8 to -32.7 ‰ in dD, with mean values of  $-5.61\pm0.04$ ‰ (2s) and  $-31.01\pm0.4$ ‰ respectively. Although low, this range is still greater than the analytical error of 0.16 ‰ and 1.4 ‰ respectively (Fig. 8b and c and Supplement S3). From July 2017 to January 2019 virtually no variability was observed in d18O and dD (Fig. 8b and c and Supplement S3). The dripwater d18O and dD in that period range 0.3 ‰ (-5.4 to -5.7 ‰ in d18O, and 2.16 ‰ (-29.2 to -31.3 ‰ in dD, with mean values of -5.61±0.05‰ (2s) and  $-30.47 \pm 0.18$ % This range is virtually at the analytical uncertainty level. By
contrast, the three drip sites with the shortest lags, WP-2, WP-3, and WP-4, exhibit higher variability in d18O and dD. In particular, sites WP-2 and WP-3 show a marked increase (0.5 ‰ in d18O between December 2016 and January 2017 (Fig. 8a), d17O varies in the same way as d18O (Supplement S4).

- PCP has already been defined Done

– see my earlier comments regarding reporting the hundredths (or thousandth!) place Done

- this correlation deserves at least a little bit of explanation. Would have been nice if the cave monitoring had included barometric pressure. . We agree that barometric pressure would have been a valuable parameter to monitor directly, and we consider this for our future work. Unfortunately, it was not logged with our available devices. The correlation is discussed in detail in the discussion section, whereas here we include only a very brief explanation. Cave air pCO2 varied from a minimum of 438 ppm in September 2016 to a maximum of 930 ppm recorded in March 2019 (Fig. 11b). Cave air pCO2 is positively correlated with cave air temperature (R2 = 0.67, p = 0.045). The highest air pCO2 values are registered when cave air temperature reaches its maximum in summer and decrease when cave air temperature is lowest in winter (see the implications in discussion Section 5.4).

– this is a long, but incomplete, sentence. "This work is aimed" Thanks for pointing this out. We adjusted this as follows: This work aims to evaluate the hydrochemical response of Waipuna Cave to environmental dynamics, and to test its suitability for speleothem-based palaeoclimate reconstructions. We explore the links between the physiochemical parameters measured in Waipuna Cave and rainfall and temperature changes at seasonal to inter-annual timescales.

- physiochemical? Done

– I am not convinced the data "confirm" anything, but they do "suggest" or "argue
**for" fracture flow Done**

– this discussion of amount effects in the rain data is too abbreviated. The origins of amount effects have been demonstrated to reflect any of a suite of drivers, including storm track, and these should be fleshed out here in more detail. Done, we added a short discussion: The distribution of the rainwater oxygen and hydrogen isotopes along the LMWL (Fig. 7) does not reveal a clear seasonal pattern. However, when comparing rainfall d180 values with the amount of precipitation across the entire monitoring period (Fig. 12, black line), we observe a positive relationship (R2 = 0.56, p =

pressure logger within and outside the cave to address questions of ventilation. As mentioned above, this parameter has unfortunately not been logged. However, the temperature difference (DT) between surface and cave air, in combination with the geomorphology of the cave, allows the characterization of ventilation. The density contrast between both air masses has long been understood as important mechanism for cave ventilation (De Freitas et al. 1982, Smithson 1991, Kowalczk & Froelich 2009). Temperature monitoring has been successfully used to investigate ventilation in several (and sometimes quite complex) caves (Breitenbach et al. 2015, Ridley et al. 2015, Riechelmann et al. 2019).

– put references in parentheses Done

– Wood Cave, located # km (direction) from Waipuna Cave, We mentioned Harrie Wood Cave because it is in a similar climate setting, however this cave is located in Australia.

– Replot some of the data from Fernandez-Cortes et al., 2008 in this paper to illustrate the effects of air pressure. We abstain from replotting the data of Fernandez-Cortes et al. because the impact of air pressure on cave ventilation has extensively been discussed elsewhere (please refer to Fairchild & Baker 2012, pages 109-114 and 122-127, and references therein, as well as the references we list in our response and the main manuscript). We use Fernandez-Cortes et al. only as reference to 'barometric caves', we adjusted the sentence to make this more clear. Although our study does not have pressure data, the model based on the changes of air density driven by changes in air masses has been shown to act in several of caves throughout the world e.g. Obir Austria (Spötl et al. 2005), Texas US (Banner et al. 2007), NE India (Breitenbach et al. 2015), Almeria Spain (Gazquez et al. 2017), and NW Germany (Riechelmann et al. 2019). Monitoring of temperature and CO2 between June 2017 and June 2018 shows that Waipuna Cave ventilation is driven by changes in the density of internal and external air in response to seasonal external temperature (Fig. 11), i.e., Waipuna Cave is a barometric cave sensu Fernandez-Cortés et al. (2008) This behavior has

HESSD
been observed in other caves globally.

– Is this section title, while poetic, a touch too flowery? The purpose of this section title is to highlight that – although in a subtle way – Waipuna cave is able to record changes in the dripwater chemistry associated to ENSO events. Waipuna Cave reacts quickly to even short and weak events such as the La Niña in summer 2017-2018. We think that the title encourages the reading and we would like to keep it as it is.

Figure 3 – please add to the figure itself the intervals for which no data are available. Don't rely solely on mentioning this in the caption. Done

References Banner J. L., Guilfoyle A., James E., Stern L. A. and Musgrove M.: Seasonal variations in modern speleothem calcite growth in Central Texas, USA. J. Sed. Res. 77, 615-622, https://doi.org/10.2110/jsr.2007.065, 2007. Breitenbach, S.F.M., Lechleitner, F.A., Meyer, H., Diengdoh, G., Mattey, D., and Marwan, N.: Cave ventilation and rainfall signals in dripwater in a monsoonal setting- a monitoring study from NE India. Chem. Geol., 402,111-124, doi.org/10.1016/j.chemgeo.2015.03.011, 2015. Fernandez-Cortes, A., Calaforra, A., and Sanchez-Martos, F.: Hydrogeochemical processes as environmental indicators in drip water: study of the Cueva del Agua (Southern Spain), Int. J. Speleol. 37, 41–52, 2008. Freitas, C. R. De., Littlbjohn, R. N., Clarkson, T. S. and Kristament, I. S.: Cave climate: Assesment of airflow and ventilation, Journal of climatology, 2, 383-397, https://doi.org/10.1002/joc.3370020408, 1982. Gázquez, F., Calaforra, J. M., Evans, N. P., and Hodell D. A.: Using stable isotopes (d17O, d18O and dD) of gypsum hydration water to ascertain the role of water condensation in the formation of subaerial gypsum speleothems. Chem. Geol. 452, 34-46, doi.org/10.1016/j.chemgeo.2017.01.021, 2017. Kowalczk, A. J. and Froelich, P. N.: Cave air ventilation and CO2 outgassing by radon-222 modelling: How fast do caves breathe?, Earth Planet. Sc. Lett., 289, 209-219, https://doi.org/10.1016/j.epsl.2009.11.010, 2010. Riechelmann, S., Breitenbach S. F. M., Schröder-Ritzrau, A., Mangini, A. and Immenhauser, A.: Ventilation and cave air pCO2 in the Bunker-Emst Cave System (NW Germany): implications for speleothem
proxy data. J. Cave Karst Stud., 81, 98-112. DOI:10.4311/2018ES0110, 2019. Ridley, H. E., Asmerom, Y., Baldini, J. U.L., Breitenbach, S. F. M., Aquino, V.V., Prufer, K. M., Culleton, B. J., Polyak, V., Lechleitner, F.V., Kennett, D. J., Zhang, M., Marwan, N., Macpherson, C. G., Baldini, L. M., Xiao, T., Peterkin, J. L., Awe, J and Haug, G. H.: Aerosol forcing of the position of the intertropical convergence zone since AD 1150, Nat. Geosci., 8, 195-200, https://doi.org/10.1038/ngeo2353, 2015. Smithson, P.A.: Inter-relationships between cave and outside air temperatures, Theoretical and Applied Climatology, 44, 65-73, https://doi.org/10.1007/BF00865553, Theor. Appl. Climatol. 1991.
Interactive comment

**Fig. 1.** Figure 3. Daily precipitation from the Waipuna meteorological station (no data is available for October and November 2018 due to instrument failure), Otorohanga Glenbrook and Te Kuiti Ews stations (da

---

## Referee Comment (RC2) · Anonymous Referee #2 · 26 Mar 2020

**Review for HESS-2019-647: "Pacific climate reflected in Waipuna Cave dripwater hydrochemistry "**

**by Nava-Fernández et al.**

**Overall:**

This manuscript aims at evaluating the hydrochemical response of the Waipuna Cave, using a 3-yr monitoring data, as a sensor of external environmental variability and, subsequently, calibrating environmentally-sensitive proxies from speleothem archives. It also attempts to put into context the cave response to ENSO influences on some relevant proxies. Overall, the work is well-written (although several punctuations are not correctly used), and well-structured (otherwise indicated in my detailed comments below). The goal is clear and the methods are adequate. Figures are nicely presented. I congratulate the authors for putting such efforts in this manuscript.

Here I provide two types of comments: (1) more of conceptual comments that may help the author provide better grounds on the main concept they want to convey, and (2) technical comments, that mainly address lines per lines comments as I found idea or expressions requiring corrections or clarifications.

**Comment type 1:**

1. **Linkage to ENSO**: I understand that you have been referring to the last three years to interpret the ENSO relationship, but to make your point more convincing, I do suggest adding a general time series with climatology (e.g., from https://climexp.knmi.nl/start.cgi) since at least 1950 along with the SO index using either Nino 1+2, Nino 3, Nino 3.4 or Nino 4 (providing a rational for why one or the other is chosen). This could potentially demonstrate the strong climate linkage of your site to ENSO (with this time series, you can indicate with color code the El Nino Event vs. La Nina Event).

2. **Classification of drip water based on flow paths:** the authors use different terminology in the classification of the drip flow throughout the text, which is quite confusing (as some terms mean the same, others are climatic based and bedrock based). It would be nicer if the authors define the nature of the flow at the beginning of their paper, provide a general classification (e.g., Type 1, Type 2, and Type 3), group the relevant literature (already in the text) to match with their own classification, and finally assign the classification with only one name: such as in line 35. This could definitely keep consistence of terminology usage throughout your work.

3. **Proxies not shown in the main text figure:** d17O (17Oexcess) is among proxies listed in the abstract and the main text, but I was surprised that it does not appear in none of the figures of the main text. If you think this could play an important role in your study, please add it in the main figure, otherwise, it should be removed from the abstract because it is a bit misleading.

**Comments type 2:**

*Line 29:* remove "," after $CO_2$

*Line 29-30:* Please indicate directly which of these measurements were continuous and which are spot measurements (so that the use of "and" in line 29 and "and/or" in line 30 are less confusing).

*Line 35:* diffuse, fracture, and combined (this order makes more sense, please re-order)

*Line 36:* how about "small" variability

*Line 37:* remove "to" after testifying

*Line 50:* The effects of both "of" these

*Line 53:* please replace "reacts" with "responds"

*Lines 53-56:* For the case of la Niña, you specified NE New Zealand, but for El Nino, you didn't. Please keep it parallel.

*Line 55:* La Nina events "bring" stronger ….

*Lines 56:* please replace "costs" with "impacts"

*Line 57:* Please replace ";" with "." And start "For example" as a new sentence.

*Note:* When I read the first paragraph of the introduction, it gave me the impression that the paper will provide longer-term records, but this is not the case. I suggest to rewrite it better to reflect well what is intended to be conveyed in the paper.

*Line 73:* your reference "see section 3.5 for definition" is not correct

*Line 75:* use "reflects" instead of "depends"

*Line 79:* Some of the key parameters…

*Line 81:* Analysis of the latter allows to "distinguish between" the processes….soil dynamics) "and the processes"….

*Lines 85-90:* I thought conduits flow group fracture and fissure flows, as primary porosity and fracture/fissure flow both reflect the nature and quality of the bedrock. If I am wrong in my understanding here, please elaborate a little bit about the conduit flow, thank you

*Line 91:* There should be a dot after variability (you may also need to explain further around this section the drip flow classification, if necessary, as it seems to play a role in your paper)

*Line 98-99:* PCP serves as a proxy system for moisture availability (Magiera et al., 2019), and affects a range of trace elements, which may either become more concentrated (increasing X/Ca) or diluted in solution (decreasing X/Ca) with increasing PCP –I am not sure I follow this sentence well. Please re-write. I think you confuse how to interpret a ratio vs. how to interpret elemental concentration.

*Line 113:* we hypothesize "that"….

*Line 113-115:* this statement is one of the reasons I suggested to the authors to add the climatology + ENSO index (as I feel that they jump the gun too quickly). Also see my comments further below..

*Line 117:* you are listing d17O here, but no data is shown in the main manuscript, but only in the supplementary. May be it is good to add such data in the main text, and provide some convincing arguments

*Line 133-136:* I think this climate information is outdated, and it is not clear from which time period is the data being reported. I suggest visiting this site (https://climexp.knmi.nl/start.cgi ) and download the relevant data using the monthly observation. From that, you can make your own climatology plot. In addition, as you'd be keen to include ENSO, I suggest plotting these climatology data with the SO index using either Nino 1+2, Nino 3, Nino 3.4 or Nino 4 (providing a rational for why one or the other is chosen).

*Line 154:* Spot cave air pCO2 were measured

*Line 155:* 10% REALLY ??(your pCO2 values are in ppm and your equipment uncertainty is 10%, something is wrong here)

*Line 161:* Ten drip sites, along with the cave stream …..

*Line 162:* if you are not going to report any results from this core at this stage, I suggest not mentioning this at all here. In fact, if you will publish it in your future work, then this paper needs to be cited. That's more logical to me

*Line 164:* terrace may not be an appropriate term here (how about platform?)

*Line 164:* Water from the cave stream was collected during each cave visit. How about rewriting this as "additional water samples from the cave stream were also collected (please indicate how many?)"

*Line 167:* when you said "previously demonstrated" would there be any reference for this?

*Line 170:* please remove the words in parenthesis

*Line 169-171:* I have suggestions for re-writing: "Drip rates at the monitored sites were determined using two independent methods. First, spot measurements were performed at all drip sites. The number of drips per minute were counted during each visit using a stop watch and counting at least 10 drips (normally at least three one-minute duration counts for the fast drip points). Second, continuous measurements were done at four drip sites (……) using automatic acoustic ….."

*Line 173-174:* cross correlation between local cumulative ….. discharge response to rainfall

Line 174-175: This is a single sentence paragraph; did you mean to have at least one more sentence here?

*Lines 186:* before and after a set of measurements of 10 to 12 samples

*Lines 192:* would like to know more about the calibration of the 17Oexcess in water, how the samples were prepared, and run?

*Line 195:* when you say "oxygen", do you refer to 18O or 17O or both, since you've been using three machines, please be specific. (if I understand well, some of the machines did not analyze 17O).

*Line 196:* "every 6 samples"  earlier you said every 10-12 samples, did I miss something?

*Line 201:* Ca and Mg are major elements, how precise were the measurements using ICP MS vs ICP-OES? (did you test this?)

*Line 215:* Please add a coma after "Glenbrook"

*Line 216:* whe**n** overlapp**ed**

*Line 218:* I am not sure if you are interpreting a daily rainfall or cumulative monthly rainfall here, please be specific

*Line 223:* replace illustrated with "shown"

*Line 235:* seepage flow, fracture flow etc… you are using different terminology in the classification of the drip flow (earlier, the classification is different), can you please provide a common classification Type 1, Type 2, Type 3, and define each and keep this consistent throughout your work?

*Line 239:* Cluster analysis using manual and logger data reveals three main groups of …

*Line 240:* Please replace the sentence "drip site…others" with "The first group isolates drip site WP-2".

*Line 240:* " A second cluster groups sites WP1-3…."

*Line 241&242:* please remove the words inside the parentheses

*Line 258:*  please use "small" instead of "low" before variability

also for the values in () you should've used the stdev of the values to make sure your statements with the analytical error are parallel.

*Line 268:* use small "l' for liters unit
it would be better to write 1000Mg/Ca and 1000Sr/Ca so it's clear (the ratios are unitless)
*Line 278:* ..chamber "recorded" between…
*Line 293-294:* I don't think it is a good idea to anticipate this statement in this paper as there is almost no data presented from that core in here.
Line 297: "free draining" -- what do you mean by this? and in which aspect?
*Line 299-303*: The use of seasonal flow vs. seepage flow in classifying the types of flow sounds a bit technically incorrect. One seems to relate to the nature of the overlying bedrock, the other to climate, which is like comparing oranges and apples. Please refer to my comments earlier (also if you'd like to include climatic classification, you could say "fracture flows are more seasonal than seepage flows", for example).
*Line 299-318:* Again, you are using a lot of technical terms to describe the nature of the flow. I'd suggest to define the nature of the flow at the beginning of your paper and assign it to only one name per category.
*Line 324-326:* please explain a bit the mechanism with regard to the light and heavy isotopes
*Line 328-329:* how could you quickly infer that?
*Line 337:* remove "(" before multi
*Line 349:* "boundary layer", what layer?
*Line 365-368:* wouldn't this reflect the amount effect?
*Line 385:* please make sure to follow the journal guidelines in using in-text citation
*Line 406:* Section 5.5: is this reference correct?
*Line 428:* please provide a subtitle that is more scientific (the current subtitle could be better for lay-audience readers, e.g., for blogs)
*Line 700:* with lag days between 11 and 16 days to
*Line 745:* there are two data points of the Feb-Aug dataset that merge with the orange data sets. Don't these data points change the linear fit? Why they are there?
*Figure S1:* what is the climatic difference between seepage flow and seasonal flow (see my comments above). I would expect that all the WPs (WP1-1, WP1-2, WP1-3, WP1-4, WP-1A, WP1B) in figure 2b should belong to one category (based on how I understand the figure 2b)
*Figure S8:* the blue diamonds seem to show a bimodal distribution: one that seems to be parallel with the orange plot, and the other detached from it. Does this represent something else?

---

## Author Comment (AC2) · 1 May 2020

Dear Referee,

We would like to thank you for your positive response, the helpful comments and for your time to help improve our manuscript. Your review is highly appreciated. Below, we answer your comments in detail.

Comment type 1: 1. Linkage to ENSO: I understand that you have been referring to the last three years to interpret the ENSO relationship, but to make your point more convincing, I do suggest adding a general time series with climatology (e.g., from https://climexp.knmi.nl/start.cgi) since at least 1950 along with the SO index using either Nino 1+2, Nino 3, Nino 3.4 or Nino 4 (providing a rational for why one or the other

[Figure]

is chosen). This could potentially demonstrate the strong climate linkage of your site to ENSO (with this time series, you can indicate with color code the El Nino Event vs. La Nina Event).

RESPONSE: We have correlated monthly rainfall anomalies with the Nino 3.4 index and the SOI index using the available station near Waipuna Cave. Rainfall data from New Plymouth station (1950-2004) shows a negative correlation with Nino 3.4 for the months June to November (see figure 1a), indicating that below average austral winter rainfall is recorded in New Plymouth during El Niño events. Comparison with the SOI index shows a fairly strong correlation between SOI and rainfall amount in the austral winter, that is, from July to November (see figure 1b). High SOI values in these months, are indicative of La Niña events, and go in hand with high precipitation over the west coast of the North Island of New Zealand. Prolonged periods of negative (positive) SOI values coincide with abnormally warm (cold) ocean waters across the eastern tropical Pacific, typical of El Niño (La Niña) episodes (https://www.ncdc.noaa.gov/teleconnections/enso/indicators/soi/).

This information has been added to the introduction: The link between ENSO events and the climate in the west coast of North Island of New Zealand (i.e. New Plymouth rainfall dataset 1950-2004) is reflected in the correlation with Nino 3.4 and SOI indices. Rainfall in this area is negatively correlated with Nino 3.4 for the months June to November, indicating that during El Niño events New Plymouth receives below average rainfall in austral winter. The SOI index shows a fairly strong correlation with austral winter rainfall (July to November). During times of positive SOI values, indicative of La Niña, higher than normal precipitation is observed at the west coast (https://climexp.knmi.nl/start.cgi).

We have also added a new reference (Ummenhofer et al., 2007) who present supporting evidence of ENSO effects over New Zealand. During El Niño events the North Island experiences drier conditions while during La Niña events the west coast of the North Island of New Zealand receives above average precipitation. We abstain from

adding another figure to the manuscript as it already has 15 figures.

2. Classification of drip water based on flow paths: the authors use different terminology in the classification of the drip flow throughout the text, which is quite confusing (as some terms mean the same, others are climatic based and bedrock based). It would be nicer if the authors define the nature of the flow at the beginning of their paper, provide a general classification (e.g., Type 1, Type 2, and Type 3), group the relevant literature (already in the text) to match with their own classification, and finally assign the classification with only one name: such as in line 35. This could definitely keep consistence of terminology usage throughout your work.

RESPONSE: Thanks for this useful suggestion, we have incorporated a simplified classification based on the three different types of flow paths we identified in Waipuna Cave and applied these throughout the manuscript to make it clearer. The first mention of this classification is in the abstract as follows:

Based on the drip response dynamics to rainfall and other characteristics we identified three types of discharge associated with hydrological routing in Waipuna Cave: i) type 1: diffuse flow, ii) type 2: fracture flow, and iii) type 3: combined flow.

RESPONSE: In the results section we have re-arranged the text to make it more fluent and define the three different drip groups:

4.2 Waipuna Cave hydrology All drip sites were hydrologically active during the monitoring period, with variable mean discharges between 10.5 and 22.1 mL s-1. The CV of the drip sites varied between 31 and 149 % (Table 1). Cross-correlation analysis between antecedent cumulative rainfall and drip rate time series from the acoustic drip loggers show different lag times for each drip. These are 19 days for WP 1-1, 15 days for WP 1-2, 16 days for WP 1-3, and 4 days for drip point WP-2 (Table 1, Fig. 5). For the drip sites where drip rates were only measured manually during the cave visits, the observed lags were 18 days for WP 1-4 and WP 1A, 11 days for WP 1B and WP-3, and 6 days for WP-4. Cluster analysis using manual and logger data reveals
three main groups of drip sites based on 25 observations of discharge at each drip site with 4 common data points among them (Fig. 6). Based on the cluster analysis we identified three flow types defined hereafter as Type 1, which includes drip sites with the slowest response to rainfall (WP 1-1, WP 1-2, WP 1-4 and WP-4); Type 2, which isolates drip WP-2 with the fastest response to rainfall; and Type 3, which includes drip sites WP 1-3 and WP-3 with intermediate response time to rainfall. For comparison we have also located the drip sites in the classification grid of Smart and Friederich (1987) (Supplement figure S1), which will be discussed in section 5.1.

RESPONSE: In the discussion section the new text reads: 5.1 Waipuna Cave hydrology Our results indicate that the monitored drip sites respond to three different infiltration pathways: i) Type 1 via diffuse, ii) Type 2 via fracture, and iii) Type 3 via combined flow. Type 1 drips WP1-1, WP 1-2, WP 1-4 and WP 1A show the slowest response to rainfall (lagging between 11 and 19 days) and belong to the Organ Loft curtain, which strongly supports the hypothesis that these sites are hydrologically connected to each other. Given that the curtain is part of the continuum from the ceiling to the floor (ca. 6 m height), it is likely that all its drip sites are mainly fed via diffuse flow through the limestone matrix, which is a function of the primary porosity of the karst (Bradley et al., 2010). Type 2 is represented by drip sites WP-2, and WP-4, which are located in the upper gallery of the Organ Loft with a greater height of the ceiling. These drips have a faster response (4 to 6 days) to antecedent rainfall, suggesting that these drips are controlled mainly by fracture flow. This is consistent with the clear identification of zones of structural weakness along the ceiling (physically representing fault or joint-like structures) and the shorter vadose flow path at these locations. The cross-correlation of the antecedent rainfall and the drip rates agrees with the cluster output for all drip sites except drip site WP-4, which has a response time of 4 days but clusters with the group of drip sites with a lag of 18 to 24 days. This can be explained by the limited size of the dataset: the drip rates of WP-4 were measured only manually, thus limiting the input for the cluster analysis compared to drip sites monitored with loggers. Finally, we grouped WP 1-3 and WP-3 into flow type 3, because these two drip sites have

similar intermediate response rates to rainfall (11 days), independent of their location in the Organ Loft chamber which is in the ceiling of the upper gallery for WP-3 and the flowstone curtain. It is likely that these drips are fed by a combination of fracture and matrix flow (Mahmud et al., 2018). Although the response time varies from days to 2–3 weeks, all drip sites forming stalagmites and feeding the flowstone reflect precipitation dynamics at sub-annual scale. The three types of drip discharge in Waipuna Cave do not satisfactorily fit into the classification model of Smart and Friederich (1987) in which drip sites WP 1A, WP 1B and WP-3 fall into the seepage flow range, while sites WP 1-1, WP 1-2, WP 1-3, WP 1-4, WP FB, WP-4, and WP-2 fall into the fracture flow range (Supplement figure S1).

RESPONSE: In the conclusion part we added: Based on geochemical and drip rate data, we identify three distinct infiltration pathways for the studied drip sites. These are Type 1: diffuse flow, Type 2: fracture flow, and Type 3: combined flow, with lagged responses to antecedent rainfall of 24–18 days, 4–6 days, and 11 days, respectively. Waipuna Cave thus quickly reacts (within less than one month) to external precipitation variability and is sensitive to sub-seasonal changes in epikarst hydrology.

3. Proxies not shown in the main text figure: d17O (17Oexcess) is among proxies listed in the abstract and the main text, but I was surprised that it does not appear in none of the figures of the main text. If you think this could play an important role in your study, please add it in the main figure, otherwise, it should be removed from the abstract because it is a bit misleading.

RESPONSE: We decided not to show d17O in a figure in the main text because it shows the same variability as d18O as can be seen in fig. S2 in the supplement. However, we agree that adding former figure S5 (now Fig. 9 in the manuscript, see figure 2 on this response) to the discussion of the 17Oexcess results in the main text improves the manuscript.

The 17Oexcess signal has been discussed before, we only adjusted the figure references: Waipuna Cave dripwaters do not show significant variations in 17Oexcess over time and most results overlap within analytical errors (Fig. 9). Recent investigations into triple oxygen isotopes in mid-latitude rainfall have reported seasonal oscillations in 17Oexcess that have been attributed to changes in relative humidity at the moisture source (i.e., where the water vapour originates), or to swings between different moisture sources with evaporation occurring under different environmental conditions (Affolter et al., 2015; Uechi and Uemura, 2019). Unlike d-excess, 17Oexcess in rainfall is apparently almost exclusively controlled by relative humidity at the water-vapour boundary layer, with insignificant temperature effects (Luz and Barkan, 2010). Thus, if there are seasonal changes in the dominant moisture source and origin of storms for the Waikato area, these would be likely to affect local precipitation and this variability could be recorded in Waipuna Cave dripwater. However, no significant variations in 17Oexcess are found over the studied period (September 2017 to October 2018, Fig. 9), suggesting that the isotope values of meteoric water are homogenized in the epikarst, and that Waipuna Cave dripwater 17Oexcess is insensitive to (sub-) seasonal changes. We suspect that, as with d18O ratios, the interannual response of cave dripwater might be controlled by long-term variations in the 17Oexcess of rainfall and changes in the relative importance of ENSO and the southern Westerlies. However, given the narrow range of 17Oexcess in rainwater in the mid-latitudes (normally < 30 per meg, Luz and Barkan, 2010) and the relatively large errors of current analytical methods (i.e., ∼ 8 per meg), longer (i.e. multi-decadal) dripwater monitoring is required to test this hypothesis.

Line 29: remove "," after CO2

RESPONSE: Done

Line 29-30: Please indicate directly which of these measurements were continuous and which are spot measurements (so that the use of "and" in line 29 and "and/or" in line 30 are less confusing).

RESPONSE: Done. The revised text now it reads as follows:

Dripwater from 10 drip sites was collected at roughly monthly intervals for a period of ca. 3 years for isotopes (d18O, dD, d-excess, d17O, 17Oexcess) and elemental (Mg/Ca, Sr/Ca) analysis. The monitoring included spot measurements of drip rates, and cave air CO2 concentration. Cave air temperature and drip rates were continuously recorded by automatic loggers. These datasets were compared to surface air temperature, rainfall, and potential evaporation from nearby meteorological stations to test the degree of signal transfer and expression of surface environmental conditions in Waipuna Cave hydrochemistry.

Line 35: diffuse, fracture, and combined (this order makes more sense, please re-order)

RESPONSE: Done

Line 36: how about "small" variability

RESPONSE: Done

Line 37: remove "to" after testifying

RESPONSE: Done

Line 50: The effects of both "of" these

RESPONSE: Done

Line 53: please replace "reacts" with "responds"

RESPONSE: Done

Lines 53-56: For the case of la Niña, you specified NE New Zealand, but for El Nino, you didn't. Please keep it parallel.

RESPONSE: Thank you for pointing this out. We added another reference (Ummen-hofer et al. 2007) in support of our claim and adjusted the text as follows:

During El Niño events, New Zealand is susceptible to increases in the frequency and intensity of westerly and southwesterly winds, accompanied by decreased rainfall in the North Island (Ummenhofer et al., 2007). In contrast, La Niña events are accompanied by stronger northeasterly winds and increased rainfall in the North Island (Griffiths, 2007; Ummenhofer et al., 2007).

Line 55: La Nina events "bring" stronger . . ..

RESPONSE: Done

Lines 56: please replace "costs" with "impacts"

RESPONSE: Done

Line 57: Please replace ";" with "." And start "For example" as a new sentence.

RESPONSE: Done

Note: When I read the first paragraph of the introduction, it gave me the impression that the paper will provide longer-term records, but this is not the case. I suggest to rewrite it better to reflect well what is intended to be conveyed in the paper.

RESPONSE: We adjusted this paragraph to better introduce our study and added a sentence at the end of this paragraph to connect the general background with the proxy reconstructions. Also, the next two paragraphs clarify this issue since we have organized the introduction from the general climatic settings of the study site to the focused objectives of our study. The adjusted text now it reads as follows:

The study of the long-term natural variability of New Zealand hydroclimate (and emergent teleconnection patterns) is a priority, both because of the effects of ENSO variability on local economic conditions and because records from this region are sparse but vital for improving the robustness of model projections of future ENSO conditions. Prior to building robust palaeoclimate (ENSO) reconstructions (e.g. by using speleothems), field sites that are highly susceptible to ENSO-related environmental changes must be

identified through monitoring campaigns.

Line 73: your reference "see section 3.5 for definition" is not correct

RESPONSE: Thanks for pointing this out. We have corrected this to section is 3.4.

Line 75: use "reflects" instead of "depends"

RESPONSE: Done

Line 79: Some of the key parameters. . .

RESPONSE: Done

Line 81: Analysis of the latter allows to "distinguish between" the processes. . ..soil dynamics) "and the processes". . ..

RESPONSE: Done

Lines 85-90: I thought conduits flow group fracture and fissure flows, as primary porosity and fracture/fissure flow both reflect the nature and quality of the bedrock. If I am wrong in my understanding here, please elaborate a little bit about the conduit flow, thank you

RESPONSE: To explain the physical karst properties, we follow the hydrogeological definition by Ford and Williams (2007), which is commonly used in the speleothem science community: "For the sedimentologists the primary porosity is created during the deposition of the rock and secondary porosity as that resulting later for diagenesis. However, for the hydrogeologist all types of bulk rock porosity are primary (sometimes called as matrix porosity), and only fracture (or fissure) porosity arising from rock folding and faulting should be considered to be secondary. When dissolution along penetrable fissures by circulating ground waters develops some pathways into pipes (conduits or caves) this is referred to tertiary porosity." Ford and Williams, 2007. We have improved the description of the main flow paths to their porosity levels:

The physical proprieties of the karst zone define the different levels of porosity (Ford and Williams, 2007). Primary porosity is a matrix of inter-granular pore space, secondary porosity is associated with joints and fractures, and tertiary porosity with solution-enhanced conduits. Seepage water experiences one or a combination of these different porosity types, which determine hydrological pathways (Fairchild and Baker 2012).

Line 91: There should be a dot after variability (you may also need to explain further around this section the drip flow classification, if necessary, as it seems to play a role in your paper)

RESPONSE: Done. We have defined the drip classification throughout the manuscript.

Line 98-99: PCP serves as a proxy system for moisture availability (Magiera et al., 2019), and affects a range of trace elements, which may either become more concentrated (increasing X/Ca) or diluted in solution (decreasing X/Ca) with increasing PCP – I am not sure I follow this sentence well. Please re-write. I think you confuse how to interpret a ratio vs. how to interpret elemental concentration.

RESPONSE: Thank you for pointing this out, we rewrote this paragraph to make it clearer:

PCP serves as a proxy system for moisture availability (Magiera et al., 2019), as it controls the distribution of trace elements in the infiltrating water during precipitation of carbonate depending on their partition coefficient prior to arrival at a stalagmite. During the dry season when the epikarst is less water-filled, PCP can occur, resulting in an increase in X/Ca ratios in solution (and subsequently speleothem), while during the wet season, when the epikarst is refilled, PCP is suppressed and dripwater X/Ca ratios are lowered (Fairchild et al. 2009).

Line 113: we hypothesize "that"….

RESPONSE: Done

Line 113-115: this statement is one of the reasons I suggested to the authors to add the climatology + ENSO index (as I feel that they jump the gun too quickly). Also see my comments further below..

RESPONSE: We adjusted the introduction with regard to ENSO and its impact on the north island of New Zealand (see our responses above) and we believe that this mitigates the statement here. Here we formulate our working hypothesis (which is well founded in previous studies) and our study is aimed at testing this hypothesis (as we explain in the next sentences). Thus, we argue that the statement is well placed, and we leave it as is.

Line 117: you are listing d17O here, but no data is shown in the main manuscript, but only in the supplementary. May be it is good to add such data in the main text, and provide some convincing arguments

RESPONSE: Since in our dataset d17O behaves very similar to d18O (see fig. S2 in the supplement) we do not discuss it in detail in the main text.

Line 133-136: I think this climate information is outdated, and it is not clear from which time period is the data being reported. I suggest visiting this site (https://climexp.knmi.nl/start.cgi ) and download the relevant data using the monthly observation. From that, you can make your own climatology plot. In addition, as you'd be keen to include ENSO, I suggest plotting these climatology data with the SO index using either Nino 1+2, Nino 3, Nino 3.4 or Nino 4 (providing a rational for why one or the other is chosen).

RESPONSE: We have updated this information using data for a period from 1950 to 2000 average data of Te Kuiti High School meteorological station from the National Climate Database CliFlo-NIWA because the climexp.knmi.nl/start.cgi site does not have information of stations in or close to the Waitomo region. The new paragraph now reads: Based on data from Te Kuiti High School, a meteorological station in the Waitomo district, the average conditions during the period from 1950 to 2000 AD include

annual rainfall of 1539 mm without any distinct rainy season. Summer and winter monthly precipitation means are similar (95 mm and 150 mm, respectively), with highest values in austral winter (July) and lowest values during austral summer (February). The mean annual temperature is 13.4°C, ranging from an average of 18.5°C in summer to 8°C in winter.

Line 154: Spot cave air pCO2 were measured

RESPONSE: Done

Line 155: 10% REALLY ??(your pCO2 values are in ppm and your equipment uncertainty is 10%, something is wrong here)

RESPONSE: Thanks for pointing this out. That was a mistake and the correct number is 1%, we corrected this as follow:

Discrete cave air pCO2 measurements were conducted during each visit in the sampling chamber using a Vaisala M170 GMP 343 Carbocap CO2 probe with a precision of $\pm 1$ %.

Line 161: Ten drip sites, along with the cave stream …...

RESPONSE: Done

Line 162: if you are not going to report any results from this core at this stage, I suggest not mentioning this at all here. In fact, if you will publish it in your future work, then this paper needs to be cited. That's more logical to me

RESPONSE: Done. The information of the WP-15-1 flowstone core has been removed.

Line 164: terrace may not be an appropriate term here (how about platform?)

RESPONSE: We edited the sentence which now reads:

Three further drip sites (WP-2, WP-3 and WP-4) are located a few meters apart below active stalactites (Fig. 2c) on an elevated section within the same chamber, and higher

in the ceiling than the first seven drip sites.

Line 164: Water from the cave stream was collected during each cave visit. How about rewriting this as "additional water samples from the cave stream were also collected (please indicate how many?)"

RESPONSE: Done. The sentence which now reads:

Additional water samples from the cave stream were also collected (one per visit). Dripwater samples for stable isotope analysis (d17O, d18O and dD) were collected and stored in sterile 2 ml or 10 ml polypropylene bottles, filled with no head space and sealed using laboratory film.

Line 167: when you said "previously demonstrated" would there be any reference for this?

RESPONSE: Done

Water samples for trace element analysis were collected in 15 ml polypropylene (Falcon) tubes, previously demonstrated to have low metal blanks (Hartland et al., 2015).

Line 170: please remove the words in parenthesis

RESPONSE: Done

Line 169-171: I have suggestions for re-writing: "Drip rates at the monitored sites were determined using two independent methods. First, spot measurements were performed at all drip sites. The number of drips per minute were counted during each visit using a stop watch and counting at least 10 drips (normally at least three one-minute duration counts for the fast drip points). Second, continuous measurements were done at four drip sites (. . . . . .) using automatic acoustic . . .."

RESPONSE: Done, thank you for the suggestion

Line 173-174: cross correlation between local cumulative . . ... discharge response to

rainfall

RESPONSE: Done

Line 174-175: This is a single sentence paragraph; did you mean to have at least one more sentence here?

RESPONSE: Thanks for pointing this out. Taking also the suggestion of the editor we have added an extra subsection at the end of the methods. In this new section all the used data analyses are described and we believe the last two sentences of the paragraph you mentioned above fit better in this new subsection. The new subsection now reads:

3.6 Data analysis In order to characterize the hydrological behavior of the drip sites the coefficient of variation (CV) (Smart and Friedrich, 1987; Baldini et al., 2006) was calculated for discharge relative to the time of data collection (CV % = ïĄş/Åń x 100, with ïĄş being the standard deviation, and Åń the mean). Cross-correlation analysis between cumulative antecedent rainfall and drip rates was used to identify the response time of the drip sites to the rainfall amount during the monitoring period. Cluster analysis was employed to classify the drips sites according to hydrological similarities. Linear regressions were used to visualise the relationships between dripwater Mg/Ca and Sr/Ca ratios, the cave air $CO_2$ and cave air temperature, as well as between rainfall amount and water isotopes. From these analyses a determination coefficient $R^2$ and p values were provided. $p < 0.05$ were considered to be statistically significant.

Lines 186: before and after a set of measurements of 10 to 12 samples

RESPONSE: Done

Lines 192: would like to know more about the calibration of the 17Oexcess in water, how the samples were prepared, and run?

RESPONSE: Done, no extra sample treatment is required for 17Oexcess measurements with the used instrumentation. We adjusted the device specifications to make it

clear:

Samples collected from March 2018 to February 2019 were measured for d17O, d18O and dD on a Picarro L2140-i at the Department of Biology and Geology at the Universidad de Almería, Spain. This instrument permits measurement of the triple oxygen and hydrogen isotope composition of liquid water with no sample pretreatment. The CRDS devices were interfaced with an A0211 high-precision vaporizer.

Line 195: when you say "oxygen", do you refer to 18O or 17O or both, since you've been using three machines, please be specific. (if I understand well, some of the machines did not analyse 17O).

RESPONSE: Here oxygen refers to both d17O and d18O, that's why we did not specify. The precision in the three machines is similar and already given at the end of the paragraph.

The long-term precision for the d-excess parameter (dD-8*d18O) was ± 0.7 ‰ on the Picarro L1102-i and ± 0.3 ‰ on the Picarro L2140-i. The long-term precision for 17Oexcess was ± 8 per meg. The calibrated value of BOTTY was indistinguishable within analytical errors when using the three different instruments, suggesting results are comparable.

Line 196: "every 6 samples" earlier you said every 10-12 samples, did I miss something?

RESPONSE: 10-20 samples in the number of samples that are bracketed by a standard, analysed to normalize to VSMOW, i.e. calibration to the international standard. Later, "every 6 samples" refers to the standard utilized for drift correction and thus evaluate the long-term precision.

Line 201: Ca and Mg are major elements, how precise were the measurements using ICP MS vs ICP-OES? (did you test this?)

RESPONSE: This is a misunderstanding; ICP-OES has not been used and all the

elements were analysed by ICP-MS (though on two instruments) and thus we did not compare these two methods. The precision of both instruments is similar, and it is now mentioned in the text:

Elemental and major cation concentrations in cave stream, drip- and rainwater were measured on two generations of instruments at the University of Waikato. Samples collected between August 2016 and October 2017 were analysed using a Perkin Elmer Elan quadrupole ICP-MS, and samples collected between November 2017 and February 2019 were analysed with an Agilent 8900 triple quadrupole ICP-MS at the Waikato Environmental Geochemistry Laboratory. The precision of both instruments is similar and all relative standard deviations (RSDs) were <5%. The ICP-MS was optimized to maximum sensitivity daily, ensuring oxides and double-charged species were less than 2 %. External calibration standards were prepared using a IV71-A multi element standard from 0.1 to 500 ppb for trace elements and single element standards were used to prepare calibration standards for major elements Ca, Fe, Si, P, S, K, Na. An internal standard containing Sc, Ge, Te, Ir, and Rh was used for all samples. Check standards were analysed every 20 samples and re-calibration was 210 performed every 100 samples. Blank samples were analysed every 10 samples to ensure minimal carryover between analyses.

Line 215: Please add a coma after "Glenbrook"

RESPONSE: Done

Line 216: when overlapped

RESPONSE: Done

Line 218: I am not sure if you are interpreting a daily rainfall or cumulative monthly rainfall here, please be specific

RESPONSE: Done. Thanks for pointing this out. Daily rainfall is correct, and it is now incorporated in the main text.

Line 223: replace illustrated with "shown"

RESPONSE: Done

Line 235: seepage flow, fracture flow etc. . . you are using different terminology in the classification of the drip flow (earlier, the classification is different), can you please provide a common classification Type 1, Type 2, Type 3, and define each and keep this consistent throughout your work?

RESPONSE: Done

Line 239: Cluster analysis using manual and logger data reveals three main groups of . . .

RESPONSE: Done

Line 240: Please replace the sentence "drip site. . .others" with "The first group isolates drip site WP-2".

RESPONSE: Done

Line 240: "A second cluster groups sites WP1-3. . .."

RESPONSE: Done

Line 241&242: please remove the words inside the parentheses

RESPONSE: Done

Line 258: please use "small" instead of "low" before variability also for the values in () you should've used the stdev of the values to make sure your statements with the analytical error are parallel.

RESPONSE: Done. We refrain from using "small variability" and keep "low variability" instead, as "small" refers to a certain range, but "low" to the change (i.e. variability). We have changed the beginning of the paragraph, also following to the comment of the first referee and considering the stdev. It now reads as follow:

Between September 2016 and June 2017, drip sites WP 1-1, WP 1-2, WP 1-3, WP 1-4, WP 1A, and WP 1B showed low variability in dripwater d18O and dD, with a range of 0.5 ‰ (-5.4 to -5.9 ‰ in d18O and 3.9 ‰ (-28.8 to -32.7 ‰ in dD, with mean values of -5.61±0.04 ‰ (2s) and -31.01±0.4 ‰ respectively. Although small, this range is still greater than the analytical error of 0.16 ‰ and 1.4 ‰ respectively (Figs. 8b, c and S3). From July 2017 to January 2019 virtually no variability was observed in d18O and dD (Figs. 8b, c and S3). The dripwater d18O and dD in that period have a range of 0.3 ‰ (-5.4 to -5.7 ‰ in d18O, and 2.16 ‰ (-29.2 to -31.3 ‰ in dD, with mean values of -5.61±0.05 ‰ (2s) and -30.47±0.18 ‰Ṫhis range is virtually at the analytical uncertainty level.

Line 268: use small "l' for liters unit it would be better to write 1000Mg/Ca and 1000Sr/Ca so it's clear (the ratios are unitless)

RESPONSE: We keep the notation as is. Liter is a non-SI unit and both L and l can be used. (https://physics.nist.gov/cuu/Units/outside.html, https://usma.org/correct-si-metric-usage).

Line 278: ..chamber "recorded" between… RESPONSE: Done

Line 293-294: I don't think it is a good idea to anticipate this statement in this paper as there is almost no data presented from that core in here.

RESPONSE: We would like to inform the reader about ongoing work and highlight the importance of the present study in a longer-term framework. Therefore, we keep an adjusted statement to this end:

This work aims to evaluate the hydrochemical response of Waipuna Cave to environmental dynamics, and to test its suitability for speleothem-based palaeoclimate reconstructions. We explore the links between the physiochemical parameters measured in Waipuna Cave and rainfall and temperature changes at seasonal to inter-annual timescales. Our results show that Waipuna Cave reflects the external environmental

dynamics on inter-annual timescales. The results and interpretation of monitoring data constitute a solid platform for the interpretation of speleothem-based reconstructions that are currently under construction.

Line 297: "free draining" – what do you mean by this? and in which aspect?

RESPONSE: Free draining refers to the soil, allowing fast drainage of infiltrating water.

Line 299-303: The use of seasonal flow vs. seepage flow in classifying the types of flow sounds a bit technically incorrect. One seems to relate to the nature of the overlying bedrock, the other to climate, which is like comparing oranges and apples. Please refer to my comments earlier (also if you'd like to include climatic classification, you could say "fracture flows are more seasonal than seepage flows", for example).

RESPONSE: Done. We have adjusted the manuscript (see our replies above) and applied three categories for the different flow routes now. When the terms seasonal flow and seepage flow are mentioned in the text these are referring to the Smart and Friederich (1987) classification. This is just a classical nomenclature used in the development of the first conceptual models of cave dripwater hydrology. We use it in this work for comparison with our own classification which is already defined.

Line 299-318: Again, you are using a lot of technical terms to describe the nature of the flow. I'd suggest to define the nature of the flow at the beginning of your paper and assign it to only one name per category.

RESPONSE: Done. See above, we have applied categories, type 1, type 2, and type 3 for the different flow routes and used it throughout the manuscript.

Line 324-326: please explain a bit the mechanism with regard to the light and heavy isotopes

RESPONSE: Done. We have added more information after the comments of the first referee and the manuscript now reads as follows:

**HESSD**

The distribution of the rainwater oxygen and hydrogen isotopes along the LMWL (Fig. 7) does not reveal a clear seasonal pattern. However, when comparing rainfall d18O values with the amount of precipitation across the entire monitoring period (Fig. 12, black line), we observe a positive relationship (R2 = 0.56, p = <0.0001). The strongest correlations between rainfall amount and d18O values are observed in austral spring and summer (R2 = 0.68 p = 0.0002, and R2 = 0.89 p = 0.0001, respectively) when temperature is highest in the Waikato area (Fig. 12, green and orange lines). Among the various climatic and geographical effects on the isotopic composition of rainwater, the 'amount effect' has been shown to significantly influence rainwater d18O in sub-tropical regions. The amount effect is the empirical negative correlation between rainfall amount and rainwater d18O (Dansgaard 1964), which arises from the partial re-evaporation and thus isotopic enrichment of rain droplets falling through relatively dry air below cloud level during periods of reduced precipitation (Dansgaard, 1964; Risi et al., 2008; Lachniet 2009; Breitenbach et al. 2010). This process affects the isotopic signature in rainfall observed in the Waitomo region in spring and summer where the correlation of decreased rain amount and lighter isotopic composition of oxygen is stronger, but not during the winter season when re-evaporation from falling rain is minimal due to high relative humidity. This is reflected in lower R2-values samples from April to September, (Fig. 13). In the wet season rain droplets are less affected by re-evaporation and remain unaltered with respect to ïĄď'18O. The seasonally contrasting isotope signatures govern the empirical amount effect (Breitenbach et al. 2010). These observations suggest that regional atmospheric conditions, associated with ENSO dynamics or strength of the Westerlies, can impose their signature on the isotopic composition of precipitation.

Line 328-329: how could you quickly infer that?

RESPONSE: This inference is based on the fact that the ENSO-associated drought events affect the isotopic composition of the rain during these events. However, as indicated in the text it is a preliminary observation suggested by the data.

[Figure]

Line 337: remove "(" before multi

RESPONSE: Done

Line 349: "boundary layer", what layer?

RESPONSE: We refer to the water-vapour interface. We have now changed this in the manuscript:

Unlike d-excess, 17Oexcess in rainfall is apparently almost exclusively controlled by relative humidity at the water-vapour boundary layer (i.e. the interface between water and free atmosphere), with insignificant temperature effects (Luz and Barkan, 2010).

Line 365-368: wouldn't this reflect the amount effect?

RESPONSE: We abstain from discussing this signal as related to the amount effect given that we have such a small dataset. While we cannot exclude the possibility that lower d18O values could be related to increased rainfall (through a more humid air column) it could also simply be related to a slightly different moisture transport history.

Line 385: please make sure to follow the journal guidelines in using in-text citation

RESPONSE: Done

Line 406: Section 5.5: is this reference correct?

RESPONSE: Yes, it is correct

Line 428: please provide a subtitle that is more scientific (the current subtitle could be better for lay-audience readers, e.g., for blogs)

RESPONSE: Done. We changed it to "ENSO signature in Waipuna Cave"

Line 700: with lag days between 11 and 16 days to

RESPONSE: Done

Line 745: there are two data points of the Feb-Aug dataset that merge with the orange

data sets. Don't these data points change the linear fit? Why they are there?

RESPONSE: There are a few sample points that fall on the orange dataset, rather than the blue, as you correctly observe. These points are explained by the different lags of the monitored drips to antecedent rainfall. While 5 of the 8 samples taken on the 7th of February 2018 fall in the blue group, indicating that they are less affected by PCP, 3 fall on the orange group, which is more strongly affected by PCP. The different chemistry in the samples of this day suggests that the cave hydrology indeed reacts within a few days to weeks to changes in infiltration. We added a sentence in the main text.

The discussion reads now: A plot of Mg/Ca versus Sr/Ca ratios displays two clusters, each along a clear trend (Figs. 9b and 14). Orange data points indicate all samples collected between October 2016 and February 2019, minus the period that comprises the blue group of samples. Blue-coloured symbols represent samples collected between February 2018 and the end of August 2018, a period with above-average rainfall, likely related to a La Niña event that developed in December 2017. Some water samples collected on 7th February 2018 fall into the same range as the orange (stronger PCP) group, while others collected on the same day, fall into the range of the blue group, supporting our notion that the cave's hydrology reacts within days to weeks (depending on drip lag response) to infiltration changes.

Figure S1: what is the climatic difference between seepage flow and seasonal flow (see my comments above). I would expect that all the WPs (WP1-1, WP1-2, WP1-3, WP1-4, WP-1A, WP1B) in figure 2b should belong to one category (based on how I understand the figure 2b)

RESPONSE: The labels seepage flow and seasonal flow stem from the original paper of Smart and Friederich (1987). Principally, seepage flow and seasonal flow should differ in their transfer time from the surface to the cave: seepage flow is slower, as it has to migrate through the host rock, while seasonal flow could be seen as similar to fracture flow – and is thus much faster transported from the surface down into the cave.

Concerning the mentioned drips (all being related to the organ loft flowstone curtain) we agree that one would expect the drips to behave quite similarly. However, this is not the case (as can also be seen in the clustering (Fig. 6), and it must be assumed that their feeding flow regimes differ near the ceiling, regardless of their very close arrangement. Regarding figure S1: This figure shows our monitored drips in a classical conceptual model of dripwater hydrology based on the mean drip rate and drip variability. The drips are here simply segregated by seasonal flow regime and seepage flow regime, depending on the coefficient of variation. Drip sites with significant seasonal variability are grouped in the seasonal flow regime, drips without such variability are placed in the seepage flow regime. We plotted the Waipuna Cave drips in this classification scheme with the aim of characterizing the infiltration routes. However, we realized that our drip sites do not fit into this older classification scheme, even if the drip sites WP1-1, WP1-2, WP1-3, WP1-4, WP-1A, WP1B belong to the same speleothem formation. That is why we considered other methods, such the statistic correlations and clustering, to classify and characterize the hydrological pathways. The old classification of Smart and Friederich (1987) can be regarded as insufficient and needs re-evaluation.

Figure S8: the blue diamonds seem to show a bimodal distribution: one that seems to be parallel with the orange plot, and the other detached from it. Does this represent something else?

RESPONSE: We agree – the blue data can be further subdivided in one group that falls on the same trend as most orange samples, where another group clearly follows a lower slope. The diagram highlights the strong effect of water availability on PCP, and the limited (or nil?) effect of cave ventilation on PCP (see section 5.4 in the manuscript). The data that are detached from the main (orange) distribution correspond to the months after a La Niña event which brought extra moisture to the cave site during the ventilated season. This trend suggests that this period of higher infiltration impacts on elemental dynamics, thus lowering the Sr/Ca vs Mg/Ca trend. Continued monitoring is required to test if the opposite would be true for El Niño events. We added

a short note in the figure caption, which now reads:

Mg/Ca and Sr/Ca ratios sorted by the period of reduced ventilation November-March (orange circles) and enhanced ventilation April-October (blue diamonds), showing the importance of water supply on elemental dynamics, and the minimal influence of the ventilation regime on PCP strength. The secondary group of blue samples following a lower slope are related to post-La Niña samples that received above-normal water supply and indicate reduced PCP above the cave.

References Ummenhofer, C. C. and England, M. H.: Interannual Extremes in New Zealand Precipitation Linked to Modes of Southern Hemisphere Climate Variability, Journal of Climate, 20, 5418-5440, 2007.

New Plymouth rainfall anomalies against Nino 3.4 index

New Plymouth rainfall anomalies against SOI index

**Fig. 1.** Figure 1. Correlations of monthly rainfall anomalies from New Plymouth station with: a) the Nino 3.4 index and b) the SOI index for the period of 1950-2004 (https://climexp.knmi.nl/start.cgi).

**Fig. 2.** Figure 9. Dripwater 17Oexcess time series of grouped according to the three main response lags (5, 18, 24 days) to antecedent rainfall (AR) at Otorohanga Glenbrook station (blue shading). a) drip site

---

## Referee Report (RR1)

**Review of the revised HESS-2019-647: "Pacific climate reflected in Waipuna Cave dripwater hydrochemistry " by Nava-Fernández et al.**

I am glad that the authors made a huge effort at improving the manuscript and at addressing the comments raised by myself and the other reviewer. Prior to its publication, however, I still have few requests for improvements. These are minor changes so I hope they won't take up too much time.
Best wishes

Note: the line numbers mentioned here correspond to the line numbers of the documents in which author responses and highlighted revised manuscript are combined as one file.

| Line no. | Comments |
|---|---|
| 55 | I don't see this edit in the text |
| 765 | Yes, I understood that ICP-OES has not been used, and that's the reason why I asked because major elements such as Ca and Mg are more reliable when measured with ICP-OES. It makes sense that results from two ICP-MS are identical, and that's why I wonder if ICP-OES and ICP-MS would be identical too? |
| 1031 | Cave microclimate |
| 1036 | Please replace "in order to verify" with "to assure" (because the research is still ongoing) |
| 1066 | ..both of these.. |
| 1075 | Add a figure reference after "SOI indices" (see explanation below). The reference in the last sentence of this paragraph may seem not appropriate. I understand I suggested it, to produce a more convincing time series to appreciate the link of your study site to these southern oscillations, and I still believe adding a figure (may be as a subset in Figure 1) of time series of rainfall and SOI index is very helpful here. |
| 1078-9 | Delete last sentence |
| 1093 | For all ref. with Fairchild and Baker, 2012, please add a page number (since this is a book, and hence without a page number, the referencing is quite vague) Same comment for the book of Ford and Williams, 2007 (L. 1124) |
| 1135 | moisture availability sounds very strange, how about rainfall? |
| 1179 | "without any distinct rainy season": earlier in the intro, you mentioned seasonality of rainfall, so I am confused with this statement |
| 1180-1 | so you think that 95 and 150 mm of rainfall is the same??? |
| 1201 | I am still not convinced with this, because 1% equals 1000ppm, if you measure near atmospheric $pCO_2$ (~400ppm), how reliable would that be? Makes me think of the previous values 10% you provided that it may indicate the upper range of $pCO_2$ that the logger can measure? |
| 1202 | Replace conducted with "done" |
| 1253 | remove the dash (drip can be a word) |
| 1286 | It may be helpful to add in the map of figure 1 the location of these weather stations. Also some other locations (L.1353) |
| 1295 | For some drip sites, where drip rates were only…. |
| 1355 | This work aims at evaluating.. |
| 1360 | ..reconstructions that are ongoing. |
| 1366 | "free draining nature" ?? how about permeability? |
| 1368 | Remove all the "via" |
| 1374 | …Organ Loft with higher ceiling. |
| 1404 | remove dot after humidity |
| 1425 | How about relative humidity inside the cave? |
| 1459 | Should this affect 17Oexcess as RH is reduced? |
| 1470 | Harrie Wood Cave (may be add the lat/long of this cave, or at least the specific region in Australia where it belongs?) |
| 1496 | $pCO_2$ instead of $CO_2$ |
| 1499 | would like to see a reference to this statement |
| 1520 | Modern ENSO signature in WC? |
| 1535 | "supporting our notion" → potentially explaining that… |
| 1584 | don't forget to replace the texts with an accessible link |